# Vascular endothelial-derived SPARCL1 exacerbates viral pneumonia through pro-inflammatory macrophage activation

Gan Zhao [1,2,3] ✉, Maria E. Gentile[1,2,3], Lulu Xue [4], Christopher V. Cosgriff[5], Aaron I. Weiner [1,2,3], Stephanie Adams-Tzivelekidis[1,2,3], Joanna Wong[1,2,3], Xinyuan Li[1,2,3], Sara Kass-Gergi[2,3,6], Nicolas P. Holcomb[1,2,3], Maria C. Basal [3,6], Kathleen M. Stewart[3,6], Joseph D. Planer[3,6], Edward Cantu[3,7], Jason D. Christie[3,6], Maria M. Crespo[3,6], Michael J. Mitchell [4], Nuala J. Meyer [3,6] & Andrew E. Vaughan [1,2,3] ✉

Inflammation induced by lung infection is a double-edged sword, moderating both anti-viral and immune pathogenesis effects; the mechanism of the latter is not fully understood. Previous studies suggest the vasculature is involved in tissue injury. Here, we report that expression of Sparcl1, a secreted matricellular protein, is upregulated in pulmonary capillary endothelial cells (EC) during influenza-induced lung injury. Endothelial overexpression of SPARCL1 promotes detrimental lung inflammation, with SPARCL1 inducing 'M1-like' macrophages and related pro-inflammatory cytokines, while SPARCL1 deletion alleviates these effects. Mechanistically, SPARCL1 functions through TLR4 on macrophages in vitro, while TLR4 inhibition in vivo ameliorates excessive inflammation caused by endothelial Sparcl1 overexpression. Finally, SPARCL1 expression is increased in lung ECs from COVID-19 patients when compared with healthy donors, while fatal COVID-19 correlates with higher circulating SPARCL1 protein levels in the plasma. Our results thus implicate SPARCL1 as a potential prognosis biomarker for deadly COVID-19 pneumonia and as a therapeutic target for taming hyperinflammation in pneumonia.

Respiratory viral pathogens such as H1N1 and H5N1 influenza and SARS/SARS-CoV-2 can destroy alveolar epithelium both by direct infection and indirectly via cytokine release from infected cells, especially by type I and type III interferons[1,2]. In some patients, this results in diffuse alveolar damage, impaired gas exchange, and ultimately, acute respiratory distress syndrome (ARDS), which bears a mortality rate of >40%[3–5]. Upon infection, activated immune cells release various cytokines, and although an appropriate inflammatory cytokine environment facilitates the recruitment of immune cells for pathogen clearance and alveolar regeneration[6,7], excessive accumulation of these cytokines can increase vascular permeability, induce additional cell death, and ultimately exacerbate lung injury[8,9]. This "cytokine storm", similar to cytokine release syndrome seen after some CAR-T therapies, ultimately manifests as sepsis, a life-threatening

[1]Department of Biomedical Sciences, School of Veterinary Medicine, University of Pennsylvania, Philadelphia, PA 19104, USA. [2]Institute for Regenerative Medicine, University of Pennsylvania, Philadelphia, PA 19104, USA. [3]Penn-CHOP Lung Biology Institute, University of Pennsylvania, Philadelphia, PA 19104, USA. [4]Department of Bioengineering, University of Pennsylvania, Philadelphia, PA 19104, USA. [5]Pulmonary and Critical Care Unit, Massachusetts General Hospital and Harvard Medical School, Boston, MA 02114, USA. [6]Division of Pulmonary, Allergy and Critical Care, Department of Medicine, Perelman School of Medicine, University of Pennsylvania, Philadelphia, PA 19104, USA. [7]Division of Cardiovascular Surgery, Department of Surgery, Perelman School of Medicine, University of Pennsylvania, Philadelphia, PA, USA. ✉e-mail: zhaogan@vet.upenn.edu; andrewva@vet.upenn.edu

complication that can present far out of proportion to the initial infection[10–12]. Cytokine-blocking strategies have been proposed to alleviate excessive inflammation, which may benefit virus-induced ARDS patients[13,14], though clinical trials targeting various cytokines have often yielded underwhelming results. Given that morbidity and mortality from viral pneumonia are associated with excessive inflammation but strategies to control this remain limited, continued research is necessary to maintain the beneficial aspects of immune responses to viral agents while limiting unnecessary tissue damage.

Macrophages are the most abundant immune cells in the healthy lung and constitute the first line of defense of the respiratory system by recognizing and engulfing pathogens, releasing cytokines, and later, promoting tissue repair, making them a critical arm of the innate immune system[15]. Macrophages exhibit remarkable phenotypic plasticity, adapting to microenvironmental cues and transforming into distinct phenotypes with specific functions. Activated macrophages are typically classified into two categories, M1 and M2, though these terms are somewhat controversial given widespread recognition of the fact that macrophage activation occurs along a spectrum rather than into discrete subtypes. M1 macrophages are defined as macrophages that produce pro-inflammatory cytokines and mediate resistance to intracellular pathogens, but these also lead to tissue destruction. M2 macrophages are, in turn, involved in anti-inflammatory responses and tissue repair/remodeling[16]. Altering the activation state of macrophages to better control the inflammatory environment represents a promising strategy for the treatment of various diseases.

The state of macrophage activation is regulated by a complex set of signals. Accumulated evidence indicates that endothelial cells (ECs) lining the pulmonary vasculature regulate lung function not only through oxygen and nutrient delivery but also participate in alveolar regeneration, immune responses, and fibrotic remodeling through the production and release of paracrine signals, also known as angiocrine factors[17–21]. Prior work has identified EC-derived signals that alter macrophage activation state to protect the lungs from injury[22,23]. Secreted protein acidic and rich in cysteine-like protein 1 (SPARCL1) is a matricellular protein reported to inhibit angiogenesis in colorectal carcinoma but also contribute to nonalcoholic steatohepatitis progression in mice[24,25]. Whether SPARCL1 is involved in lung inflammation or contributes to viral pneumonia progression is unknown.

Here, we demonstrate lung capillary ECs adopt a distinct transcriptomic and phenotypic state upon viral injury to generate high levels of SPARCL1, which in turn acts through TLR4 to promote a pro-inflammatory M1-like state in macrophages, exacerbating inflammation and increasing the severity of viral pneumonia. Further, we show that antagonism of TLR4 can specifically rescue morbidity in SPARCL1 overexpressing mice, suggesting a potential therapeutic intervention for pneumonia patients with high levels of circulating SPARCL1.

## Results

### Dynamic endothelial transcriptomics reveals increased Sparcl1 expression after influenza injury

Pulmonary gas exchange restoration after viral pneumonia requires vascular repair to restore the heterogeneous assemblage of lung endothelial cells (ECs)[9,26–28]. To explore the dynamics of endothelial subpopulations during regeneration after viral lung injury, mouse lung ECs were isolated (CD45⁻EpCAM⁻CD31⁺) by FACS on day 0 (D0, uninjured), day 20 (D20), and day 30 (D30) post influenza infection, and subsets/clusters were then identified by single-cell transcriptomic profiling (Fig. 1A). Based on the well-characterized EC subset signature genes[29,30], we identified six EC clusters (Supplementary Fig. 1A, B), including lymphatic ECs (Prox1), venous ECs (Bst1), arterial ECs (Gja5), proliferating ECs (Mki67), and 2 capillary EC clusters, aerocytes (aCap, Car4,) and general capillary ECs (gCap, Gpihbp1). Further analysis of these EC subtypes revealed 2 lymphatic endothelial subsets, lymphatic ECs_01 (Ccl21aʰⁱ lym_ECs, signature genes: Ccl21a, Mmrna, and Nts) and

lymphatic ECs_02 (Prox1ʰⁱ lym_ECs, signature genes: Prox1, Sned1, and Stab1), and 3 gCap subsets, gCap ECs_01 (Dev.ECs, signature genes: Hpgd, Tmem100, and Atf3), gCap ECs_02 (Immu.EC, signature genes: Cd74, Sparcl1 and Cxcl12) and gCap ECs_03 (Gm26917, Nckap5, and Syne1) (Supplementary Fig. 1C). GO (gene ontology) pathway enrichment analysis revealed gCap ECs_01 and gCap ECs_02 genes were enriched in vascular developmental and immune regulatory pathways, respectively (Supplementary Fig. 1D), confirming the identification of gCap ECs, Dev.ECs (devEC) and Immu.ECs (immuneEC), recently reported by Zhang et al.[28].

Interestingly, we found that the two subgroups of gCap ECs, Dev.ECs and Immu.ECs, exhibited dynamic changes during injury, with Dev.ECs significantly reduced during infection (D20) and gradually returning to baseline levels after recovery from pneumonia (D30). Immu.ECs, on the other hand, showed the opposite trend (Fig. 1B). Largely absent in uninjured lungs, Immu.ECs mainly appeared after injury, indicating a potential role in vascular responses to injury. Pseudotime analysis of these two gCap_EC subclusters suggests that Immu.ECs may originate from Dev.ECs (Supplementary Fig. 1E). Therefore, we focused on transcriptomic changes in these two gCap EC subtypes (Fig. 1C). SPARCL1, a matricellular protein reported to inhibit angiogenesis in colorectal carcinoma[25], was broadly expressed in Immu.ECs and significantly increased in ECs after injury (Fig. 1D–F). Immunostaining showed that SPARCL1 was mainly expressed in capillary ECs (especially gCap ECs), as well as mesenchymal/stromal cells (Supplementary Fig. 2A–F), and was significantly increased on day 20 after influenza injury (Fig. 1G). In healthy lungs, SPARCL1 was mainly found in the mesenchymal cells, with minimal presence in ECs, as confirmed by intracellular flow cytometry (Supplementary Fig. 2G and H). However, during influenza pneumonia, its expression markedly increases. Notably, we observed elevated SPARCL1 expression in ECs up to 60 days after influenza infection, despite a slight decrease from peak levels around day 15 (Supplementary Fig. 2I–K). The above results were recapitulated in subsequent quantitative polymerase chain reaction (qPCR) analysis for Sparcl1 mRNA in isolated ECs and enzyme-linked immunosorbent assay (ELISA) analysis for SPARCL1 protein in bronchoalveolar lavage fluid (BALF) and serum (Fig. 1H, I and Supplementary Fig. 2L, M). Interestingly, at a very late stage of injury, 60 days post-infection, we observed high levels of SPARCL1 concentration only in the serum (compared to the uninjured condition), while in BALF, it nearly returned to baseline levels (Fig. 2K, L). This suggests that vascular endothelium is the primary source of SPARCL1 in alveoli, and the completion of vascular repair prevents endothelial-derived SPARCL1 from leaking into the alveolar space. Taken together, these results demonstrate a subpopulation of gCap EC (Immu.ECs) expressing very high levels of Sparcl1 appears during injury.

### Endothelial ablation of Sparcl1 mitigates influenza-induced lung injury

Given that the injury-induced population of Immu.ECs were characterized by high expression of Sparcl1, we proceed to further probe the function of this gene. To explore the role of endothelial SPARCL1 in the pathogenesis of viral pneumonia, we crossed VECadᶜʳᵉᴱᴿᵀ² mice with novel Sparcl1ᶠˡᵒˣ mice. We used a CRISPR-Cas9 strategy in mouse embryonic stem cells (Supplementary Fig. 3A and B) to generate homozygous mutant mice and, upon crossing, ultimately proceeded with selective ablation of Sparcl1 in ECs of adult mice (referred to as ECˢᵖᵃʳᶜˡ¹⁻ᴷᴼ) via tamoxifen administration. Sparcl1ᶠˡᵒˣ/ᶠˡᵒˣ mice lacking Cre (referred to as WT) were used as the control group. The qPCR analysis on sorted lung ECs from uninjured mice confirmed the deletion of Sparcl1 (Supplementary Fig. 3C), and ELISA data further validated a significant reduction in SPARCL1 protein levels in BALF and lung tissue homogenate on day 12 after influenza infection. However, no significant differences ($p > 0.05$) were observed under homeostasis, providing further confirmation that the primary source of SPARCL1 in

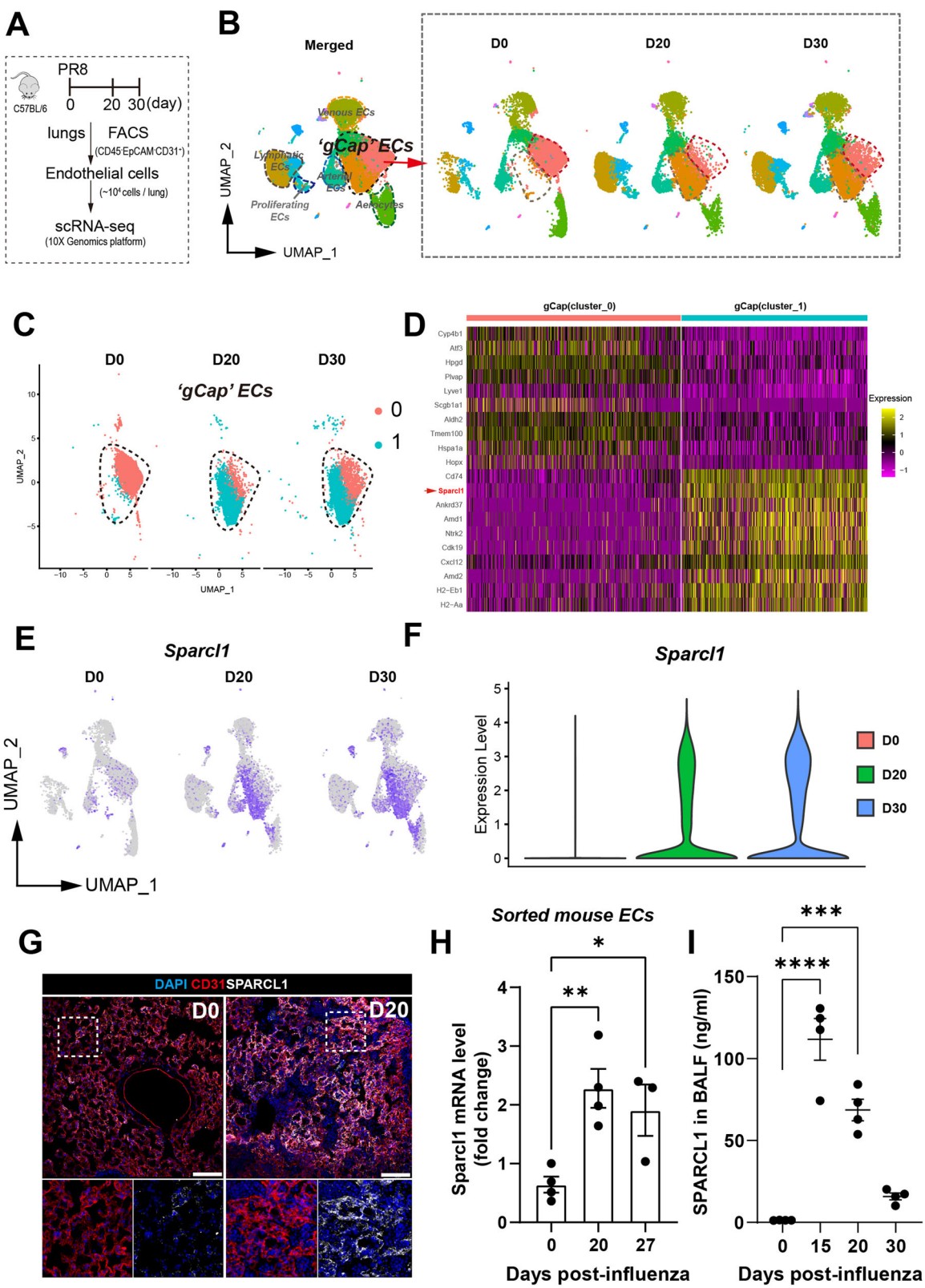

alveoli is from vascular endothelium (Supplementary Fig. 3D and E). Intriguingly, $EC^{Sparcl1-KO}$ mice demonstrated less severe pneumonia symptoms, evidenced by less initial body weight loss and faster recovery to baseline levels (Fig. 2A) as well as improved capillary oxygen saturation compared to WT mice (Fig. 2B). Additionally, $EC^{Sparcl1-KO}$ mice trended toward higher probability of survival (~75%) compared to WT (less than 50%) by day 27 post infection when

challenged with a high dose of influenza virus (Fig. 2C). We next ana-lyzed local inflammatory cytokines associated with viral pneumonia, including respiratory distress induced by both influenza and SARS-CoV-2[11,31,32]. Cytokine levels of TNF and IL-6 in the BALF were normal during homeostasis. However, during influenza pneumonia, $EC^{Sparcl1-KO}$ mice exhibited lower levels of proinflammatory cytokines/factors/ chemokines, including TNF, IL-6, IFN-γ, CXCL-1, TIMP1, C5a, CXCL-10

**Fig. 1 | Single-cell transcriptomics reveals transcriptional dynamics in gCap ECs and increased SPARCL1 expression after viral injury. A** Schematic of mouse lung endothelial single-cell sequencing preparation. **B** ScRNA-seq analysis for mouse lung ECs sorted from uninjured (D0) and on 20 and 30 days after influenza infection (marked as D20 and D30, respectively). Uniform manifold approximation and projection (UMAP) plots showing the dynamics in gCap ECs. **C** The 2 gCap EC clusters of interest, cluster_0 (Dev.ECs) and cluster_1 (Immu.ECs), were subsetted from (**B**). **D** Heatmap showing the top 20 differentially expressed genes of cluster_0 (Dev.ECs) and cluster_1(Immu.ECs). **E** UMAP analysis reveals that *Sparcl1* is predominantly expressed in gCap ECs, especially in Immu.ECs. **F** Violin plots showing *Sparcl1* expression level in mouse lung ECs sorted from D0, D20 and D30, respectively. **G** Representative immunostaining of SPARCL1 in endothelial cells (CD31) in both uninjured (D0) and D20 after influenza infection lung tissues. Scale bar, 100 μm. **H** qPCR analysis of *Sparcl1* in isolated lung ECs (CD45⁻CD31⁺) sorted on days 0 (uninjured), 20 and 27 after influenza infection, $n = 3–4$ mice per group (each dot represents one mouse), independent biological replicates. D0 vs. D20: $p = 0.006$; D0 vs. D27: $p = 0.044$. **I** The concentration of SPARCL1 in bronchoalveolar lavage fluid (BALF) was measured by ELISA at 0 (uninjured), 15, 20, and 30 days after influenza infection, $n = 4$ mice per group, independent biological replicates. D0 vs. D15: $p < 0.0001$; D0 vs. D20: $p = 0.0001$. Data in **H** and **I** are presented as means ± SEM, calculated using one-way analysis of variance (ANOVA), followed by Dunnett's multiple comparison test. *$P < 0.05$, **$P < 0.01$, ***$P < 0.001$ and ****$P < 0.0001$. Source data are provided as a Source Data file.

and BLC (CXCL-13) (Fig. 2D–F and Supplementary Fig. 3F–H), but a trend toward increased levels of the anti-inflammatory cytokines IL-4 and IL-10 in the BALF on day 12 after influenza infection compared with WT mice (Fig. 2G and H). Moreover, we also observed decreased total protein and cells in the BALF of $EC^{Sparcl1-KO}$ mice on day 12 post-infection (Fig. 2I and J). Notably, deficiency in endothelial Sparcl1 reduced the levels of the classical monocyte chemoattract CCL2 (also known as MCP-1) in BALF (Fig. 2K) and serum (not significant, Supplementary Fig. 3I) during infection (day 12). This directed our attention to myeloid populations, especially macrophages and monocytes.

We did not observe obvious changes in the number of macrophages during homeostasis (Supplementary Fig. 3J) after endothelial deletion of *Sparcl1* but observed a reduction in total macrophages during influenza infection (Fig. 2L). To assess whether this reduction resulted from impaired recruitment, dendritic cells (DCs) and monocytes were further analyzed. Both are crucial for orchestrating immunity to respiratory virus infection[33–36], and we observed a slight, though not significant, reduction in inflammatory monocytes (iMonocytes, Ly6C⁺CD11b⁺) with endothelial deletion of *Sparcl1*. This may be associated with a decrease in local MCP-1, and no significant differences were observed in DCs, including inflammatory DCs (iDCs, Ly6C⁺ DCs) (Supplementary Fig. 4A and B). Moreover, selective expression of ligands or chemokines in myeloid cells impacts the activity of other immune cell populations. For example, selective expression of CXCL-9 and PD-L1 in monocytes can influence the recruitment or activation of T cells[36]. We thus focused on CXCL-9- or PD-L1-expressing myeloid cells, including iMonocytes, DCs, as well as alveolar macrophages (AMs) and interstitial macrophages (IMs). We did not observe obvious alterations in this CXCL9 or PD-L1 expressing cells (Supplementary Fig. 5A and B). Subsequently, we assessed changes in other major immune cell populations in the lung and found that endothelial deletion of *Sparcl1* did not result in obvious changes in the number of T, B, natural killer (NK), innate lymphoid cells (ILCs), or neutrophils (Supplementary Fig. 6A, B). Moreover, loss of *Sparcl1* in ECs did not affect viral load (Supplementary Fig. 3K), suggesting the tissue repair process is affected by local inflammation levels rather than uncontrolled viral replication. These results are reinforced by our observations that mice lacking EC *Sparcl1* exhibit dampened local inflammatory tissue damage, further confirmed by histologic analysis of the $EC^{Sparcl1-KO}$ lungs (Fig. 2M and N), as judged by a previously described unbiased computational imaging approach[37]. These experiments indicate that endothelial deficiency in Sparcl1 protects against severe influenza pneumonia, at least partially by attenuating local inflammation.

**Sparcl1 overexpression worsens influenza-induced pneumonia**
To further confirm that EC-derived SPARCL1 negatively contributes to pneumonia severity, we again targeted *Sparcl1* cDNA to the ROSA26 locus, preceded by a loxP flanked "stop" sequence, in mouse ES cells. Upon generation of these mice, we then crossed these animals with the VECad$^{CreERT2}$ strain to develop conditional endothelial *Sparcl1* knock-in/overexpression mice (Fig. 3A and B), *VECad$^{CreERT2}$; Sparcl1$^{+/WT}$* or *VECad$^{CreERT2}$; Sparcl1$^{+/+}$* mice (referred to as $EC^{Sparcl1-OE}$). *VECad$^{CreERT2}$;*

*Sparcl1$^{WT/WT}$* mice (referred to as WT) were used as the control group. Upon tamoxifen administration, SPARCL1 overexpression in the lungs of $EC^{Sparcl1-OE}$ mice was confirmed by western blotting (Fig. 3C). Endothelial overexpression of *Sparcl1* did not cause phenotypic differences under uninfected conditions (PBS administration), as evaluated by body weight changes and blood oxygen saturation (Supplementary Fig. 7A and B). As expected, upon influenza infection, $EC^{Sparcl1-OE}$ mice displayed exaggerated pneumonia outcomes, as demonstrated by greater weight loss, prolonged recovery period (related to baseline level of body weight at D0) (Fig. 3D), worse lung respiratory function during pneumonia (impaired oxygen saturation) (Fig. 3E), lower probability of survival (Fig. 3F) and increased total protein and cells in the BALF (Fig. 3G and H). Moreover, overexpression of SPARCL1 exacerbates the local inflammatory response during pneumonia, which was assessed by the elevated pro-inflammatory cytokines/chemokines or factors, such as TNF, IL-1β, IL-6, IFN-γ, MIG/ CXCL-9, TIMP-1, BLC/CXCL-13 (Fig. 3I-J and Supplementary Fig. 7C–E) and decreased anti-inflammatory cytokines IL-4 and IL-10 in BALF in comparison to WT mice by day 20 (Fig. 3K and L). We also tested TNF, IL-1β and IL-6 in BALF under homeostasis and observed no significant changes, though they began to trend higher on day 10 post-infection in $EC^{Sparcl1-OE}$ mice (Fig. 3I). These data suggest that the extended physiologic recovery time observed in $EC^{Sparcl1-OE}$ mice reflects long-term inflammation. Similarly, we observed increased CCL2/MCP-1 levels in BALF with endothelial overexpression of *Sparcl1* (Fig. 3M and Supplementary Fig. 7E), but no significant changes of CCL2 in serum (Supplementary Fig. 7F), and no differences in viral load in the lungs during infection (Supplementary Fig. 7G). However, we noted increased numbers of pulmonary macrophages, especially recruited/interstitial macrophages during injury (day 20 after infection), while there were no significant differences in homeostasis between WT and $EC^{Sparcl1-OE}$ mice (Supplementary Fig. 7H). Additionally, we did not observe obvious effects on other immune cells, including T cells, B cells, NK cells, ILCs, neutrophils, iMonocytes and DCs (Supplementary Fig. 8A and B) or changes of CXCL-9- or PD-L1-expressing myeloid cells (Supplementary Fig. 8C and D). This further supports the previous conclusion that EC-derived SPARCL1 aggravates local lung inflammation in pneumonia, thus worsening lung injury.

**SPARCL1 promotes proinflammatory changes in macrophage phenotypes in vivo**
To further address the mechanisms underlying exacerbation of pneumonia outcomes attributed to endothelial Sparcl1, we assessed whether ablation or overexpression of Sparcl1 affected EC angiogenic proliferation, as predicted by reports that SPARCL1 can act as an angiostatic factor[25]. We administered the nucleoside analog 5-ethynyl-2-deoxyuridine (EdU) (50 mg/kg, intraperitoneally) and intracellular EdU flow analysis was used to quantify proliferative ECs (Supplementary Fig. 9A). Surprisingly, we observed no significant changes in EC proliferation upon either endothelial ablation (Supplementary Fig. 9A–C) or overexpression (Supplementary Fig. 9D–F) of Sparcl1. Given these results and the fact that SPARCL1 is a secreted

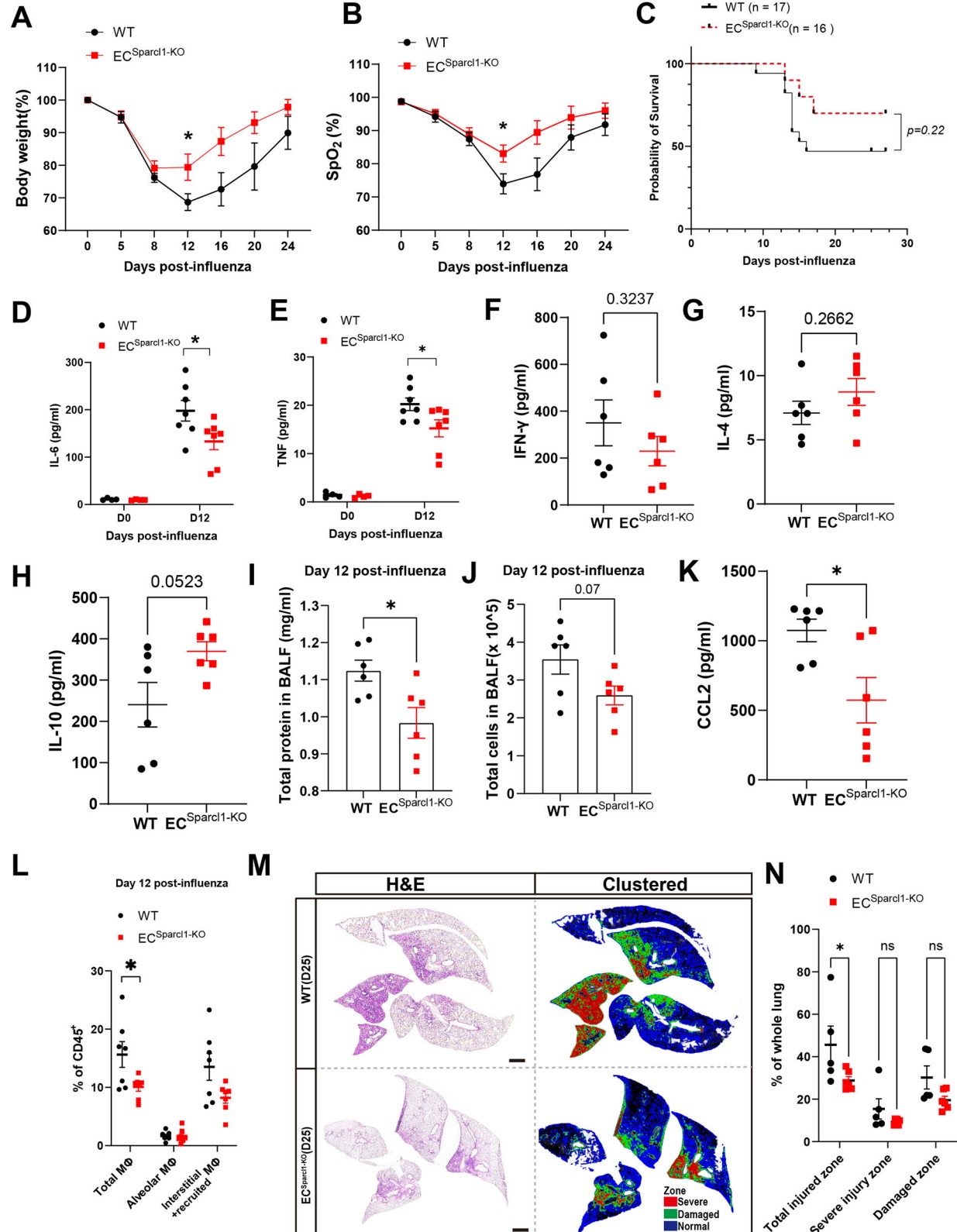

matricellular protein, we reasoned that SPARCL1 might instead be acting as a paracrine signaling molecule, influencing the phenotype of other cell types involved in lung injury and repair.

Since SPARCL1 triggered more severe inflammation upon injury and macrophage (MΦ) infiltration (Figs. 2 and 3), we first ruled out the impact of SPARCL1 on macrophage proliferation, migration, and apoptosis, though we do note a protective effect on hydrogen

peroxide ($H_2O_2$)-induced cell apoptosis in vitro (Supplementary Fig. 10A–C). Thus, we speculated that SPARCL1 promotes the release of pro-inflammatory cytokines by affecting macrophage polarization. Macrophages participate in both injury / inflammation and tissue repair by releasing cytokines and chemokines and can be subdivided into classical activated (M1) and alternatively activated (M2) macrophages, which exhibit pro- and anti-inflammatory phenotypes,

**Fig. 2 | Endothelial loss of Sparcl1 attenuates influenza-induced pneumonia.**
**A** Time course of changes in body weight and **B** capillary oxygen saturation in WT
and $EC^{Sparcl1-KO}$ mice after influenza infection, n = 7 mice per group, independent
biological replicates. **A** p = 0.04; **B** p = 0.04. **C** Kaplan–Meier survival curves after
influenza infection, log-rank test. **D–H** The levels of pro-inflammatory cytokines IL-
6 (**D**), TNF (**E**), IFN-γ(**F**), IL-4(**G**), and IL-10 (**H**) in bronchoalveolar lavage fluid (BALF)
were measured by ELISA in WT and $EC^{Sparcl1-KO}$ mice at day 0 (uninjured) and/or
D12 days after influenza infection, each dot represents one mouse, n = 4–7 mice per
group, independent biological replicates. **D** p = 0.03; **E** p = 0.04. **I** and **J** Total pro-
tein (**I**) and cells (**J**) were quantified in BALF on day 12 post influenza infection, n = 6
mice per group, **I** p = 0.018, **J** p = 0.07. **K** CCL-2 concentration in BALF from WT
(wild-type) and $EC^{Sparcl1-KO}$ mice at day 12 after infection. n = 6 mice per group,
independent biological replicates, p = 0.02. **L.** Quantification of the proportion of

total macrophages (CD64+F4/80+), alveolar macrophages (CD45+Ly6G-CD64+F4/
80+SiglecF+) and interstitial and recruited macrophages (CD45+Ly6G-CD64+F4/
80+SiglecF-) in CD45+ live cells at day 12 after influenza infection in WT (n = 7 mice)
and $EC^{Sparcl1-KO}$ (n = 6 mice) mice, independent biological replicates. Gated as shown
in Supplementary Fig. 11A. Total macrophages: WT vs. $EC^{Sparcl1-KO}$: p = 0.033. **M** Left:
tile scan images of H&E stain at 25 days post-infection, demarcated boxes indicate
different injury zones. Right: clustered injury zone maps produced from left H&E
images, scale bars, 1 mm. **N** Quantification of injured area in different injury zones in
**M**, n = 5 mice per group, independent biological replicates. Total injured zone: WT
vs. $EC^{Sparcl1-KO}$: p = 0.033. Data in **A**, **B**, **D** to (**L**) and (**N**) are presented as means ± SEM,
calculated using unpaired two-tailed t-test. Data in (**C**) were calculated using log-
rank test. *P < 0.05, ns not significant, P > 0.05. Source data are provided as a Source
Data file.

respectively. M1 macrophages are reported as the main sources of
several proinflammatory cytokines, including TNF, IL-1β and IL-6, act-
ing to amplify the inflammatory response[38,39]. Thus, we examined
whether SPARCL1 contributes to pro-inflammatory responses by
affecting macrophage M1/M2 polarization. Lung macrophages
(CD45+Ly6G-CD64+F4/80+) were subgated into SiglecF+ alveolar mac-
rophages (AMs) and SiglecF- interstitial and/or recruited macrophages
(I/RMs). Macrophages were then sub-phenotyped using the well-
described CD86 and CD206 antigens[40] to distinguish M1_like
(CD86+CD206-) and M2_like (CD86-CD206+) macrophage populations
in vivo (Supplementary Fig. 11A). We observed significantly increased
M1_like and decreased M2_like macrophage populations in $EC^{Sparcl1-OE}$
mice compared with WT mice on day 20 post influenza infection, both
in total MΦ (CD64+F4/80+) as well as specifically within AMs and I/RMs
(Fig. 4A–C, Supplementary Fig. 11A). However, EC overexpression of
Sparcl1 did not significantly alter the M1/M2 macrophage proportions
under homeostasis (Supplementary Fig. 11B), which we postulate is
due to SPARCL1 protein failing to infiltrate into the alveolar space when
the blood vessels are intact without injury. Flow cytometry analysis and
immunostaining for the M2_like macrophage marker RELMα showed a
decreased proportion of M2_like macrophages in $EC^{Sparcl1-OE}$ mice on
day 20 post-influenza infection (Fig. 4D–F), further confirming our
macrophage polarization flow cytometry data.

RNA sequencing (RNA-seq) of isolated lung macrophages
(CD45+Ly6G-CD64+F4/80+) on day 20 post-influenza infection was
used to further examine the consequences of EC overexpression of
Sparcl1. Principal components analysis (PCA) analysis was performed,
with $EC^{Sparcl1-OE}$ samples forming a tight cluster distinct from WT sam-
ples, confirming that EC Sparcl1 overexpression results in significant
transcriptional variation effects on macrophages (Fig. 4G). Gene set
enrichment analysis (GSEA) demonstrated that differentially expres-
sed genes were significantly enriched in the cytokine–cytokine
receptor signaling pathway, and specifically, EC$^{Sparcl1-OE}$ positively
regulated the expression of inflammatory pathway genes (cytokines,
receptors, etc.) (Fig. 4H). Differential gene analysis showed that typical
M1_like signature genes, including *Cd86*, *Cd80*, and *Ifng*, *Ccl3*, *Ccl4*,
*Ccl5*, etc were significantly up-regulated in $EC^{Sparcl1-OE}$ mouse macro-
phages, while M2_like signature genes, including *Arg1*, *Cd163*, *Mrc1*, and
*Chil3* were significantly down-regulated (Fig. 4I), suggesting that the
macrophage polarization switch is a direct response to endothelial
overexpression of Sparcl1. We also asked whether endothelial deletion
of Sparcl1 might inversely affect M1/M2-like macrophage polarization
during injury in comparison to Sparcl1 overexpression. Macrophages
and their M1/M2 subsets were again gated as mentioned above on day
12 post influenza infection (Supplementary Fig. 12A). In agreement
with results using $EC^{Sparcl1-OE}$ mice, EC loss of Sparcl1 resulted in fewer
M1_like but more M2_like macrophages during pneumonia (Supple-
mentary Fig. 12B–D), with no obvious changes under homeostasis
(Supplementary Fig. 12E and F). Taken together, these data support a
model wherein EC-derived SPARCL1 induces an inflammatory
response by triggering the polarization of M1-like macrophages and/or

inhibiting the transformation of M2 macrophages, thereby promoting
the persistence of inflammation that contributes to the exacerbation
of pneumonia.

## The SPARCL1-induced pro-inflammatory macrophage pheno-
type requires TLR4 signaling in vitro
To better understand the molecular basis of SPARCL1-induced M1
macrophage polarization, bone marrow (BM) cells were isolated and
differentiated into mature macrophages, BM-derived macrophages
(BMDMs), which were validated by flow cytometry analysis (CD11b+F4/
80+) (Supplementary Fig. 13A, B) and then treated with recombinant
mouse SPARCL1 protein. NF-κB acts as the critical M1 polarization reg-
ulator via transcriptional activation of pro-inflammatory cytokine gene
expression[16], and SPARCL1 significantly induced phosphorylation of NF-
κB p65 (Fig. 5A and Supplementary Fig. 14A) and the release of pro-
inflammatory cytokines TNF, IL-1β and IL-6 (Fig. 5B). To eliminate
potential endotoxin (lipopolysaccharide, LPS) contamination in the
recombinant protein, BMDMs were pre-treated with or without the LPS
inhibitor, Polymyxin B, before being exposed to SPARCL1 protein. The
results indicate that Polymyxin B effectively inhibits the phosphorylation
of NF-κB p65 induced by LPS, but this inhibitory effect is not observed in
the groups treated with SPARCL1 (Supplementary Fig. 14B). Moreover,
to further validate our findings, we utilized the *SPARCL1* overexpression
(SPARCL1-OE) human lung endothelial cell line, iMVECs[9], and trans-
duced iMVECs with an empty plasmid (WT) as a control (Supplementary
Fig. 14C and D). The supernatant from both SPARCL1-OE and WT iMVECs
was then collected to treat the human macrophage cell line, THP-1,
which was differentiated by PMA (phorbol 12-myristate 13-acetate)
(Supplementary Fig. 14E). Our data confirmed that the conditioned
medium from *SPARCL1*-OE iMVECs triggered the phosphorylation of NF-
κB p65 and the expression of pro-inflammatory cytokines *TNF* and *IL-6*
(Supplementary Fig. 14F and G). These findings strongly suggest that
SPARCL1 derived from endothelial cells directs macrophages toward a
pro-inflammatory (M1-like) phenotype. Morphologically, SPARCL1-
treated BMDMs appeared similar to BMDMs treated with LPS (a
known activator of the M1 phenotype), further indicating that SPARCL1
induces BMDMs towards an M1 pro-inflammatory phenotype (Supple-
mentary Fig. 15A). Next, we explored whether SPARCL1 enhances an
M1_like phenotype or hinders the M2_like phenotype. BMDMs were
polarized into M1/M2 macrophages by treatment with LPS (50 ng/ml) or
IL-4 (20 ng/ml) for 24 h and subsequently incubated with recombinant
SPARCL1 protein (10 μg/ml) for 24 h (Fig. 5C). Our data showed that
M2_like macrophages (CD206+CD11b+F4/80+) retain significant CD206
expression even upon withdrawal of IL-4 treatment for 24 h (M2 + PBS,
-80%), but this was significantly reduced by SPARCL1 treatment (Fig. 5D
and Supplementary Fig. 5B). Further analysis of the supernatant from
these experiments demonstrated that IL-1β and IL-6 levels were slightly
increased after SPARCL1 treatment compared with M1 control group
(Supplementary Fig. 15C), however, SPARCL1 significantly increased the
levels of TNF, IL-1β and IL-6 in comparison with M2 control (Supple-
mentary Fig. 15C). Moreover, the M2 macrophage marker genes *Mrc1*

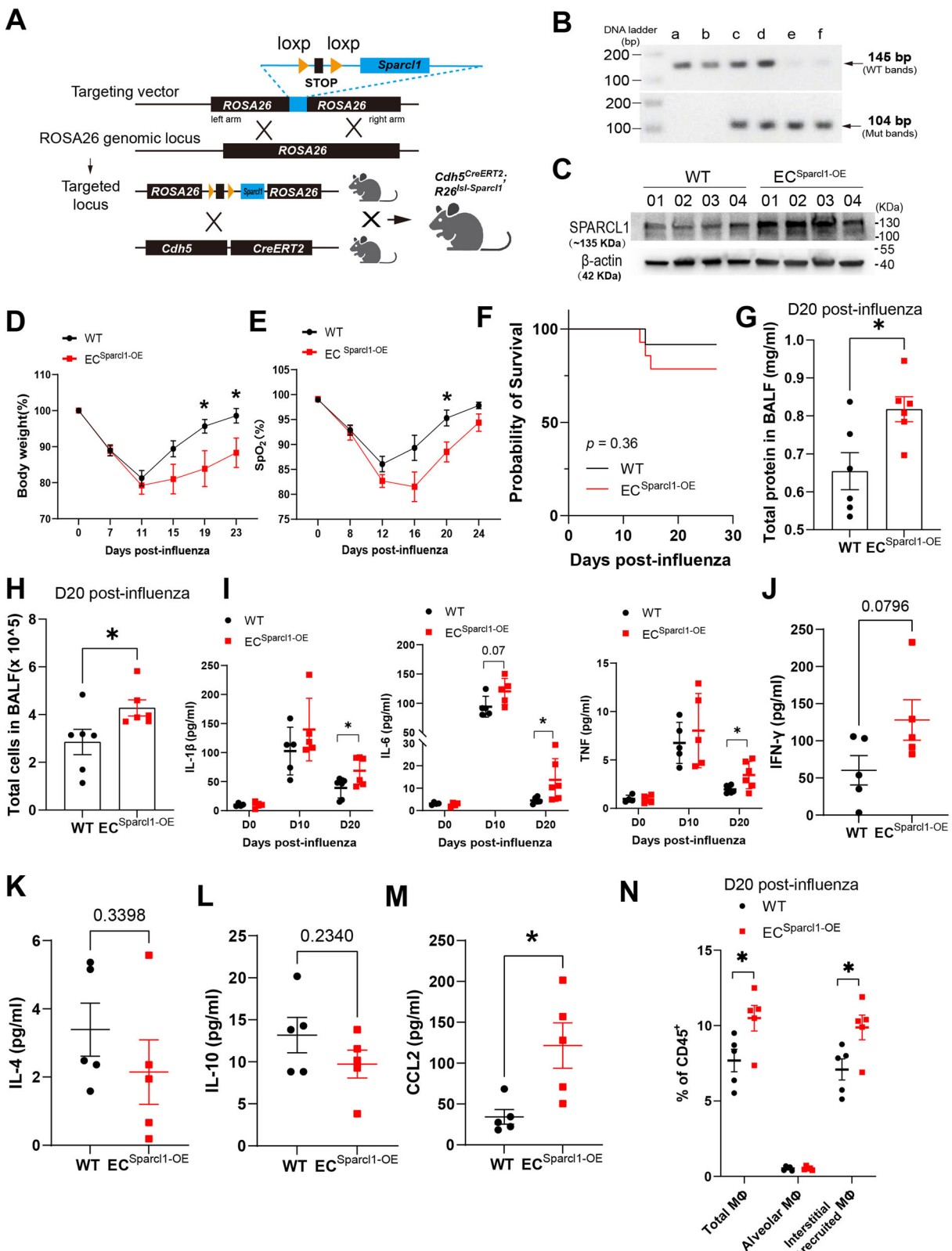

and *Chil3* were detected by qPCR and were highly expressed in the M2 control group as expected but suppressed upon subsequent SPARCL1 treatment (Fig. 5E). M2 macrophages treated with SPARCL1 were morphologically distinct from the M2 control, again exhibiting a pro-inflammatory M1 macrophage appearance (Supplementary Fig. 15D). These data indicate that SPARCL1 suppresses the M2_like phenotype and promotes a transition toward the M1 pro-inflammatory phenotype.

SPARCL1 has been reported to directly bind Toll-like receptor 4 (TLR4) in hepatocytes to activate a downstream inflammatory response cascade[24], and stimulation of TLR4 directly activates NF-κB[41]. We therefore utilized the TLR4-specific inhibitor, TAK-242 (10 μM) to treat BMDMs 1 h prior to SPARCL1 treatment (10 μg/ml). TAK-242 almost completely blocked SPARCL1-induced phosphorylation of NF-κB p65 (Fig. 5F) and downstream release of TNF, IL-1β and IL-6 (Fig. 5G).

**Fig. 3 | Endothelial overexpression of Sparcl1 exacerbates influenza-induced pneumonia. A** Schematic of the strategy used to generate Sparcl1 knock-in mice targeting ROSA26 locus. **B** DNA gel image for genotyping of endothelial Sparcl1 knock-in mice. Lanes *a* and *b* represent wild-type mice with PCR product at 145 bp. Lanes *c* and *d* represent heterozygous mice with product at 145 bp (wt) and product at 104 bp (mut). Lanes *e* and *f* represent homozygous mice with PCR product at 104 bp. **C** Western blot of SPARCL1 in whole lung tissue from *EC^{Sparcl1-OE}* (*VECad^{CreERT2}; Sparcl1^{+/ut}* or *VECad^{CreERT2}; Sparcl1^{+/+}*) or WT (*VECad^{CreERT2}*) mice 2 weeks after 5 doses of tamoxifen, the image showing representative data from n = 4 biological replicates. **D** Time course of changes in body weight and **E** capillary oxygen saturation in WT and *EC^{Sparcl1-OE}* mice after influenza infection. **D** WT: *n* = 9 mice, *EC^{Sparcl1-OE}*: *n* = 7 mice, independent biological replicates; D19: *p* = 0.02, D23: *p* = 0.03. **E** *n* = 5 mice per group, independent biological replicates; D20: *p* = 0.03. **F** Kaplan–Meier survival curves after influenza infection, log-rank test. WT: *n* = 12 mice, *EC^{Sparcl1-OE}* *n* = 14 mice. Total protein (**G**) and cells (**H**) were quantified in BALF on day 20 post influenza infection, *n* = 6 mice per group, independent biological replicates. **G** *p* = 0.02; **H** *p* = 0.046. **I** The concentration of pro-inflammatory cytokines, IL-1β,

IL-6 and TNF in BALF were measured by ELISA in WT and *EC^{Sparcl1-OE}* mice at 0 (uninjured, *n* = 3 independent biological replicates per group), 10 (*n* = 5 independent biological replicates per group) and 20 (*n* = 5 independent biological replicates per group) days post-influenza infection. IL-1β (D20): *p* = 0.037; IL-6 (D20): *p* = 0.04; TNF (D20): *p* = 0.032. ELISA measurement of IFN-γ (**J**), IL-4 (**K**), IL-10 (**L**), and CCL-2 (**M**) in WT and *EC^{Sparcl1-OE}* mice on day 20 post-infection. *n* = 5 mice per group, independent biological replicates. CCL-2: p = 0.017. **N**. Quantification of the proportion of total macrophages (CD64⁺F4/80⁺), alveolar macrophages (CD45⁺Ly6G⁻CD64⁺F4/80⁺SiglecF⁺) and interstitial and recruited macrophages (CD45⁺Ly6G⁻CD64⁺F4/80⁺SiglecF⁻) in CD45⁺ live cells at day 20 after influenza infection in WT and *EC^{Sparcl1-OE}* mice, *n* = 5 mice per group, independent biological replicates. Gated as shown in Supplementary Fig. 11A. Total macrophages: *p* = 0.037; interstitial and recruited macrophages: *p* = 0.032. Data in **D**, **E** and **G** to **H** and **J** to **N** are presented as means ± SEM, calculated using an unpaired two-tailed *t*-test. Data in **I** is presented as means ± SD, calculated using an unpaired two-tailed *t*-test. *P < 0.05. Data in **F** were calculated using log-rank test. *P < 0.05, ns: not significant, P > 0.05. Source data are provided as a Source Data file.

This was also corroborated in TLR4 knockout BMDMs, indicating that the knockout of TLR4 resulted in the inability of both LPS and SPARCL1 to induce the phosphorylation of NF-κB p65 (Supplementary Fig. 14B). GSEA revealed EC overexpression of Sparcl1 positively correlated with the TLR4 receptor pathway, and we observed upregulation of TLR4 downstream genes in *EC^{Sparcl1-OE}* mouse macrophages (Fig. 5H and Supplementary Fig. 15E). Taken together, these data strongly indicate that SPARCL1 induces an M1 proinflammatory macrophage phenotype by activating TLR4/NF-κB signaling (Fig. 5I).

### TAK-242 administration ameliorates pneumonia exacerbations induced by endothelial SPARCL1 overexpression

Having established that EC overexpression of Sparcl1 worsens viral pneumonia through TLR4 signaling, we next explored whether inhibition of TLR4 is sufficient to ameliorate influenza-induced injury in mice. To test this hypothesis, WT and *EC^{Sparcl1-OE}* mice were treated with TAK-242(3 mg/kg, i.p, every 2 days) from day 7 to 20 post influenza infection and mouse lungs were harvested on day 20 or 25 (Fig. 6A). TAK-242 treatment significantly improved pneumonia symptoms in *EC^{Sparcl1-OE}* mice as assessed by reduced weight loss, shorter recovery time (Fig. 6B and Supplementary Fig. 16A), and improved gas exchange as indicated by higher oxygen saturation (Fig. 6C and Supplementary Fig. 16B) compared with vehicle treatment. Furthermore, treatment with TAK-242 slightly reduced the number of macrophages, particularly the recruited macrophages, which were likely recruited by the endothelial overexpression of Sparcl1 (Supplementary Fig. 16C). Moreover, treatment with TAK-242 reduced M1_like and slightly increased M2_like macrophages (Fig. 6D-F) and reduced proinflammatory cytokines levels (Fig. 6G) including TNF, IL-1β and IL-6 in *EC^{Sparcl1-OE}* mice on day 20 after infection. Intriguingly, TAK-242 did not exhibit appreciable therapeutic effects in WT mice, including no significant differences in body weight loss (Supplementary Fig. 16A), oxygen saturation levels (Supplementary Fig. 16B), the number of macrophages, changes in M1/M2 macrophage populations (Supplementary Fig. 16C–E) and BALF cytokines (with the exception of TNF) (Fig. 6G). We interpret these observations to indicate that some degree of TLR4-mediated inflammatory signaling is beneficial, therapeutic targeting of TLR4 may require a strict administration time window, and inhibition of TLR4 may only be therapeutic in individuals with very high levels of SPARCL1. Taken together, targeting TLR4 signaling effectively alleviates the exacerbation of inflammation caused by overexpression of SPARCL1 in ECs, confirming that SPARCL1 mediates downstream inflammatory responses through TLR4 signaling in vivo. These findings also spur the prediction that if patients exhibit heterogeneity in SPARCL1 levels upon viral lung injury, treatment with TLR4/NF-kB inhibitors may be therapeutic for a subset of these patients with the highest SPARCL1 levels.

### High levels of SPARCL1 are associated with poor outcomes in patients with viral pneumonia

To determine the clinical relevance of our animal-based observations, lungs from COVID-19 ARDS patients (collected after viral clearance, COVID_donors) and healthy control lungs (Healthy_donors) were collected. Immunostaining for SPARCL1 revealed higher expression of SPARCL1 in vascular ECs (ERG⁺SPARCL1⁺) in post-COVID lungs compared to healthy lungs (Fig. 7A). qPCR analysis of ECs isolated from post-COVID and healthy lungs indicated that *SPARCL1* was significantly upregulated in COVID endothelium (Fig. 7B). Survival analysis demonstrated that the level of SPARCL1 in the plasma of patients with fatal COVID-19 disease was significantly higher than that of those who survived (Fig. 7C). These observations reinforce our findings in vitro and in our transgenic mouse models, indicating that high levels of SPARCL1 are closely related to the development and exacerbation of COVID-19 pneumonia, and detection of SPARCL1 in plasma represents a potential method to evaluate the prognosis of viral pneumonia as well as to potentially identify patients who may response positively to SPARCL1, TLR4, or NF-κB inhibition.

## Discussion

Endothelial cells (ECs) lining the capillary network around the alveoli play a crucial role in lung physiology, responsible for gas exchange, nutrient transport, and leukocyte trafficking. Although they are not typically directly infected by respiratory viruses, viral infections can nonetheless trigger a severe local inflammatory response leading to EC apoptosis, necrosis, and other forms of cell death[8]. ECs undergo a continuous and dynamic adjustment of their functions in response to pathogens/damage-associated molecular patterns or cytokines, leading to the classical endothelial activation phenotype characterized by induction of adhesion molecules ICAM-1, VCAM-1, and E-selectin[42]. This leads to increased adherence of leukocytes to the endothelial surface and heightened transendothelial migration, facilitating the influx of immune cells, which, in turn, contribute to both injury and repair processes in the tissue. As in other organs, lung ECs are heterogeneous, consisting of arterial, venous, lymphatic, and two distinct subsets of capillary ECs, aerocytes (aCap) ECs and general capillary (gCap) ECs, thought to govern gas exchange and capillary repair, respectively[27,29]. While various studies have investigated the differences in the responses of lung EC types to injury and found evidence of functional differences among these subpopulations, the lack of strict and reliable tracing markers has limited their conclusions, which are primarily based on bioinformatic prediction. Further experimental evidence is needed to fully validate the functional consequences of these transcriptomic differences.

In this study, using single-cell transcriptome analysis, we uncovered two subpopulations of gCap ECs that exhibited alternating

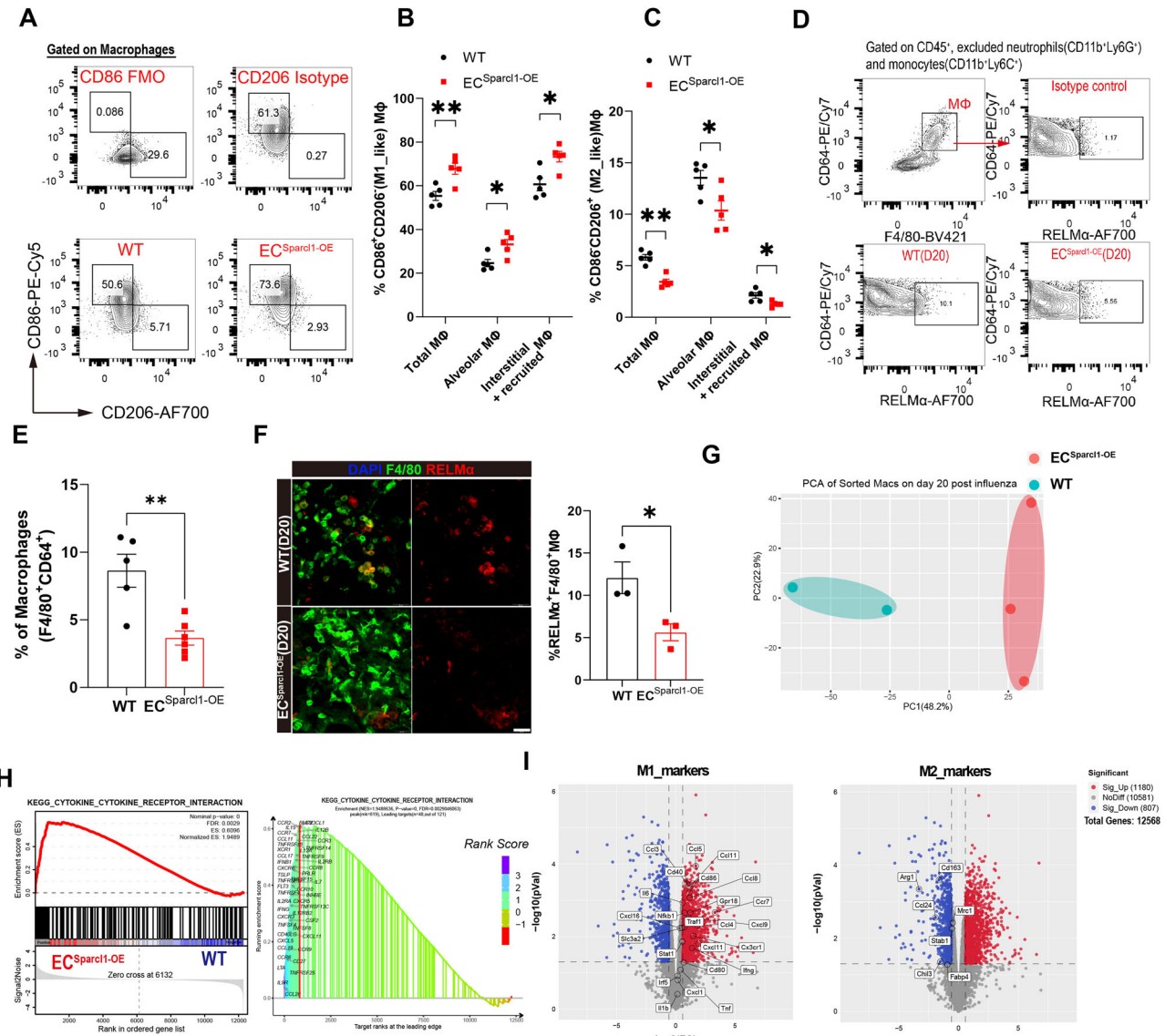

**Fig. 4 | Endothelial overexpression of Sparcl1 promotes M1-like polarization of lung macrophages. A** Representative gating scheme for identification of pulmonary M1-like (CD86$^+$CD206$^-$) and M2-like (CD86$^-$CD206$^+$) macrophages (CD45$^+$Ly6G$^-$CD64$^+$F4/80$^+$) at day 20 after influenza infection in WT and $EC^{Sparcl1-OE}$ mice. Macrophage gating strategies as shown in Supplementary Fig. 11A. Quantification of the proportion of **B** M1-like (CD86$^+$CD206$^-$) and **C** M2-like (CD86$^-$CD206$^+$) macrophages in total lung macrophages (CD45$^+$Ly6G$^-$CD64$^+$F4/80$^+$), alveolar macrophages (CD45$^+$Ly6G$^-$CD64$^+$F4/80$^+$SiglecF$^+$) and interstitial and recruited macrophages (CD45$^+$Ly6G$^-$CD64$^+$F4/80$^+$SiglecF$^-$) at day 20 after influenza infection in WT and $EC^{Sparcl1-OE}$ mice, $n = 5$ mice per group, independent biological replicates. WT vs. $EC^{Sparcl1-OE}$ mice: **B** $p = 0.005$(Total MΦ), $p = 0.015$(Alveolar MΦ), $p = 0.01$(Interstitial+ recruited MΦ); **C** $p = 0.002$(Total MΦ), $p = 0.026$(Alveolar MΦ), $p = 0.019$(Interstitial+ recruited MΦ). **D** Gating strategy for M2-like macrophage (RELMα$^+$) in WT and $EC^{Sparcl1-OE}$ mice on day 20 post influenza infection. **E** Quantification of M2-like macrophage (RELMα$^+$/CD64$^+$F4/80$^+$) by flow cytometry analysis in WT ($n = 5$ independent biological replicates) and $EC^{Sparcl1-OE}$ ($n = 6$

independent biological replicates) mice, $p = 0.0031$. **F** Left: representative immunostaining of lung M2_like macrophage (RELMα$^+$F4/80$^+$) in WT and $EC^{Sparcl1-OE}$ mice on day 20 post influenza infection, scale bars, 25 μm. Right: quantification of the proportion of M2_like macrophages (RELMα$^+$F4/80$^+$/F4/80$^+$) in left, $n = 3$ mice per group, independent biological replicates, $p = 0.038$. **G** Principal components analysis (PCA) indicates transcriptomic changes in lung macrophages between WT ($n = 2$ independent biological replicates) and $EC^{Sparcl1-OE}$ mice ($n = 3$ independent biological replicates) on day 20 post-influenza infection. **H** Gene set enrichment analysis (GSEA) of RNA-seq profiles of purified lung macrophages isolated from WT and $EC^{Sparcl1-OE}$ mice on day 20 post influenza infection. Enrichment score and $p$ value are displayed. **I** Volcano plot indicating the M1-associated genes upregulated in lung macrophage from $EC^{Sparcl1-OE}$ mice and M2-associated genes down-regulated on day 20 post influenza infection. Data in **B**, **C**, and **E** are presented as means ± SEM, and data in **F** is presented as means ± SD, calculated using an unpaired two-tailed $t$-test. *$P < 0.05$, **$P < 0.01$. Source data are provided as a Source Data file.

behavior during influenza-induced lung injury. By analyzing their characteristic genes, we found that they were similar to the two subpopulations of ECs recently described by Zhang et al.[28] so we adopted the naming conventions of Dev.ECs and Immu.ECs. Of note, we observed dramatically reduced numbers of Dev.ECs on day 20 post-influenza, but by day 30, when lung function and oxygen saturation had largely returned to normal, Dev.ECs had been largely restored,

indicating these cells likely contribute to lung homeostasis. Further analysis revealed that the Dev.ECs highly expressed *Atf3*, a known stress-induced transcriptional regulator[43], suggesting that endothelial *Atf3* may be involved in maintaining lung homeostasis, reinforced by a recent study demonstrating the importance of Atf3-expressing ECs in lung regeneration[44]. The Immu.ECs, which primarily appear after injury and may be unique to lung injury/repair, have received limited

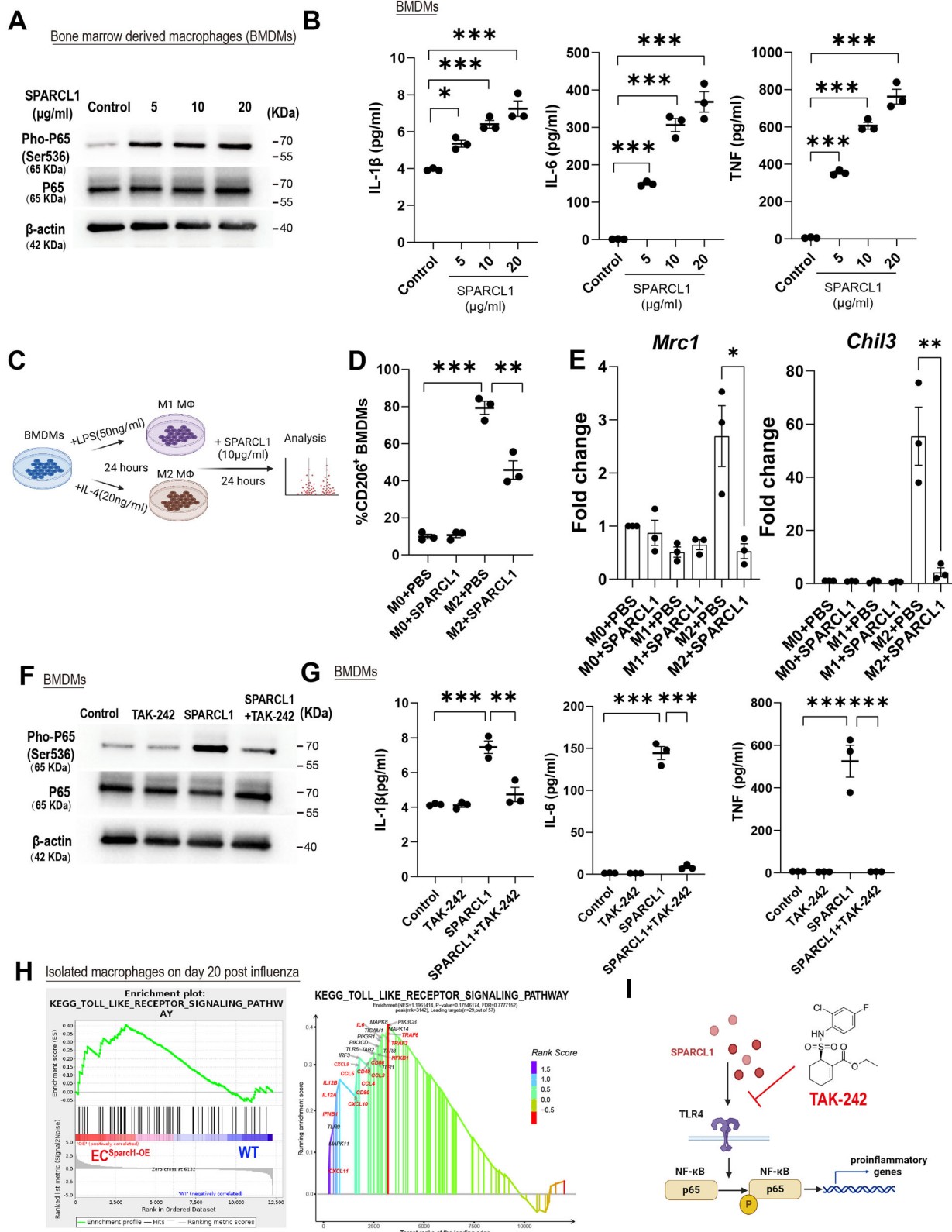

attention in previous studies. Thus, we focused on these cells. Further analysis revealed that the Immu.ECs highly expressed *Sparcl1*, and given the paucity of functional studies on this gene, we pursued further characterization.

SPARCL1, a matricellular protein and member of the SPARC protein family, has been described as a blood vessel-derived anti-angiogenic (angiostatic) protein[25]. However, overexpression or deletion of SPARCL1 in ECs in vivo did not show obvious effects on EC proliferation. SPARCL1 is also expressed in pericytes / mural cells in some tissues and is reported to be required for vascular maturation and integrity[25,45]. In the lung, however, capillary ECs (especially aerocytes) are largely discontinuous with pericytes[46,47], so we postulated that SPARCL1 expression from pericytes might be less relevant in this context. Instead of direct effects on the vasculature, our findings

**Fig. 5 | Sparcl1 induction of M1-like macrophages depends on TLR4 in vitro.**
**A** Western blotting analysis of indicated proteins in bone marrow-derived macrophages (BMDMs) treated with different doses of recombinant SPARCL1 protein (0–20 µg/ml) for 1 h, representative image from $n = 3$ biological replicates.
**B** BMDMs were treated with different doses of recombinant SPARCL1 protein (0–20 µg/ml, Sino Biological) for 24 h, and cell supernatant was collected. Pro-inflammatory cytokines IL-1β, IL-6 and TNF levels in supernatant were measured by ELISA, $n = 3$ biological replicates per group. IL-1β: $p = 0.01$ (Control vs. 5), $p = 0.0003$ (Control vs. 10), $p < 0.0001$ (Control vs. 20); IL-6: $p = 0.0005$ (Control vs. 5), $p < 0.0001$ (Control vs.10), $p < 0.0001$ (Control vs. 20); TNF: $p < 0.0001$ (Control vs. 5), $p < 0.0001$ (Control vs. 10), $p < 0.0001$ (Control vs. 20). **C** Schematic of SPARCL1 treatment of BMDMs. **D** Quantification of the proportion of M2-like (F4/80⁺CD206+) macrophages in M2 polarized BMDMs after being treated with SPARCL1 (10 µg/ml, Sino Biological) for 24 h. $n = 3$ biological replicates per group. M0 + PBS vs. M2 + PBS: $p < 0.001$; M2 + PBS vs. M2 + SPARCL1: $p = 0.003$. **E** qPCR analysis for M2 macrophage genes (*Chil3* and *Mrc1*) in M1 and M2 polarized BMDMs after being treated with SPARCL1 (10 µg/ml) for 24 h, $n = 3$ biological replicates per group. *Mrc1*:M2 + PBS vs. M2 + SPARCL1: $p = 0.021$; *Chil3*:M2 + PBS vs. M2 + SPARCL1: $p = 0.0098$. **F** BMDMs were pre-treated with TLR4 inhibitor, TAK-

242(10 µM) for 1 h and then incubated with or without SPARCL1 (10 µg/ml, Sino Biological) for 1 h. Phosphorylation of NF-κB was detected by western blot assay, $n = 3$ biological replicates per group. **G** BMDMs were pre-treated with TLR4 inhibitor, TAK-242(10 µM) for 1 h and then incubated with or without SPARCL1 (10 µg/ml, Sino Biological) for 24 h, and cell supernatant was collected. Pro-inflammatory cytokines IL-1β, IL-6 and TNF levels in supernatant were measured by ELISA, $n = 3$ biological replicates per group. IL-1β (Control vs. SPARCL1: $p = 0.0002$; SPARCL1 vs. SPARCL1 + TAK-242: $p = 0.006$); IL-6 (Control vs. SPARCL1: $p < 0.0001$; SPARCL1 vs. SPARCL1 + TAK-242: $p < 0.0001$); TNF (Control vs. SPARCL1: $p < 0.0001$; SPARCL1 vs. SPARCL1 + TAK-242: $p < 0.0001$). **H** GSEA analysis indicates that endothelial overexpression of Sparcl1 in vivo positively engaged the TLR4 signaling in isolated macrophages (see Fig. 4G–I). **I** The TLR4 inhibitor TAK-242 blocks SPARCL1-induced NF-κB activation, thereby inhibiting macrophage transition to a pro-inflammatory phenotype. Data in **B**, **D**, and **G** are presented as means ± SEM, calculated using one-way analysis of variance (ANOVA), followed by Dunnett's multiple comparison test. Data in **E** are presented as means ± SEM, calculated using unpaired two-tailed *t*-test; *$P < 0.05$, **$P < 0.01$, ***$P < 0.001$. Source data are provided as a Source Data file. Schematics and icons created with BioRender.com.

suggested that the high expression of SPARCL1 by ECs during pneumonia contributes to the worsening of lung injury by driving inflammation, ultimately causing more harm than benefit. Our ELISA data showed that SPARCL1 was significantly upregulated during pneumonia before gradually returning to normal levels during recovery, so we speculated that the high expression of SPARCL1 in influenza-induced lung injury was related to the level of local inflammation. These findings reinforce previously described studies showing that SPARCL1 activates the hepatic inflammatory response by acting as an endogenous ligand for TLR4 and contributes to liver injury in steatotic mice[24], and our work here points to macrophage activation as the major driver of this SPARCL1-mediated inflammatory exacerbation.

Macrophages play a pivotal role in orchestrating the initiation and resolution of inflammation, as well as repair responses in the lungs[48]. The diversity of macrophage function is often binned into polarized states, with M1 and M2 subtypes implicated as both drivers and regulators of disease[38]. It has been demonstrated that either the persistence of inflammatory macrophage numbers or prevention of their conversion to a reparative anti-inflammatory phenotype can further delay tissue repair following injury[49]. Therefore, identifying and characterizing the mechanisms that drive macrophages to exhibit pro or anti-inflammatory activity is critical to promote the resolution of tissue inflammation/repair responses. We show that vascular-derived SPARLC1 exacerbates pneumonia by agonizing TLR4 and promoting the polarization of pro-inflammatory macrophages. Further, we also observed that SPARCL1 converted M2-like reparative macrophages toward a more M1-like state. Intriguingly, similar effects on macrophage activation in adipose tissue have been described for SPARC[50], again acting through TLR4, suggesting the SPARC matricellular protein family may share a conserved function in modulating macrophage activity.

Though our work here elucidates novel mechanisms of endothelial-macrophage crosstalk and macrophage activation, important questions remain. While not formally demonstrated here, all indications are that Immu.ECs essentially represent an activated/inflamed state of gCaps rather than a de novo population. If true, the initiating signals driving the transcriptomic and phenotypic switch from Dev.ECs to Immu.ECs remain unknown. Although it has been demonstrated in vitro that *SPARCL1* expression can be induced by pro-inflammatory cytokines dependent upon cell confluency[25], it is difficult to know what exactly this confluency-dependent effect means in vivo, where the endothelium is already confluent at steady state. In addition, while our transgenic mouse models clearly indicate that the endothelial source of SPARCL1 is critical for the observed phenotypes, our work does not address whether there may be an additive role for

mesenchymal sources of SPARCL1, nor does it assess whether SPARC might synergize with SPARCL1 to influence macrophage behavior in the lung.

Our in vitro and in vivo experiments are consistent with previous studies[24] that demonstrate SPARCL1 activates the downstream pro-inflammatory signaling pathway NF-κB through direct binding to TLR4 and promoting M1-like polarization of macrophages. Similarly, it was reported in human osteosarcoma (OS) patients that higher SPARCL1 expression is positively correlated with M1 macrophage infiltration[51]. However, we recognize that it would be ideal to use mice with endothelial overexpression of SPARCL1 and simultaneous knockout of macrophage TLR4. However, this approach would require dual recombinases to prevent simultaneous overexpression of SPARCL1 in the myeloid lineage and TLR4 deletion in the endothelial lineage, e.g. "*Cdh5^CreERT2;loxp-stop-loxp-Sparcl1;CCR2^DreER;TLR4^fox/rox*", an approach which is not possible with currently available models. Additionally, although we have observed an association between circulating SPARCL1 and post-COVID recovery, the difficulty in obtaining sufficient numbers of samples from patients hospitalized with pure influenza pneumonia raises uncertainty about the role of SPARCL1 in the human influenza setting. As such, we recognize that SPARCL1's predictive and functional roles may differ between different etiologies of pneumonia.

While multiple studies have established the importance of resident and recruited lung macrophages in both pulmonary inflammation and tissue repair processes, much remains to be discovered regarding the underlying signaling pathways and mechanisms of multilineage paracrine communication involved. Our findings reveal that the vascular-derived molecule SPARCL1 is a critical paracrine signaling factor between capillary endothelial cells and macrophages and that macrophage responses to SPARCL1 contribute to the severity of pneumonia. Notably, the expression levels of endothelial SPARCL1 in the uninjured lung do not impact the myeloid cell population within the homeostatic lung. This may be because the intact vascular wall and alveolar structure prevent SPARCL1 from penetrating into the alveoli, a point further supported by the absence of detectable differences in SPARCL1 concentrations in uninjured BALF (Supplementary Fig. 3D). Additionally, our results indicate that SPARCL1 influences the recruitment of macrophages (Figs. 2L and 3N; Supplementary Figs. 3J and 7H) and affects their polarization, as evidenced by the levels of MCP-1/CCL-2 in the BALF (Figs. 2K and 3M). Intriguingly, CCL-2/MCP-1 is a well-known target of transcription factor NF-κB[24,52]. SPARCL1 may drive a "vicious cycle" of macrophage activation by promoting the transformation of recruited macrophages towards a pro-inflammatory state (NF-κB activation), increasing CCL-2/MCP-1 production and thus

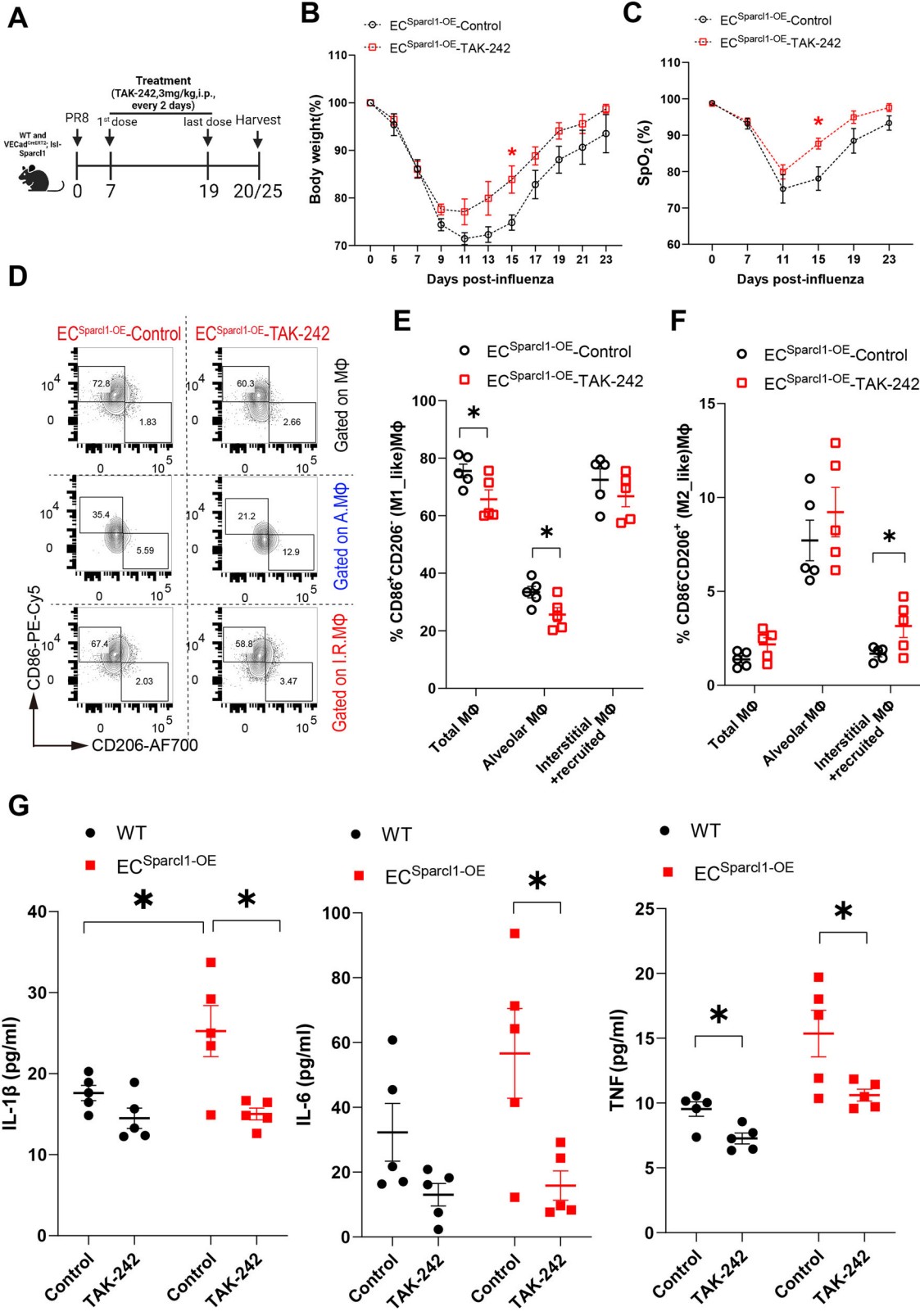

recruitment of additional monocytes/macrophages to the injury lung, thereby exacerbating the inflammatory response (Fig. 8). In this sense, SPARCL1 could be considered as a potential driving factor of pathological inflammation, and while SPARCL1 may have important tissue-protective effects as well[53], our model points to a threshold beyond which SPARCL1 causes overexuberant inflammation. Although we did not observe changes in the total numbers of other immune cells, such

as T cells, B cells, and DCs, during influenza pneumonia, we did not assess phenotypic changes within these cell types, such as regulatory and cytotoxic attributes[54,55]. However, given that viral load was unaffected by endothelial *Sparcl1* levels, differential inflammatory activation of macrophages exacerbating tissue damage appears to be the most parsimonious model to explain our observations. Integration of our findings in mouse models and patient samples leads us to suggest

**Fig. 6 | Blockade of TLR4 ameliorates the exacerbation of pneumonia induced by endothelial overexpression of Sparcl1. A** Timeline for TAK-242 administration and sampling. WT and $EC^{Sparcl1-OE}$ mice were treated with TAK-242 (3 mg/kg, i.p.) or equal volume of vehicle control (DMSO). **B** and **C** Time course of changes in (**B**) body weight and (**C**) capillary oxygen saturation in $EC^{Sparcl1-OE}$ mice treated with or without TAK-242 after influenza infection. **B** $n = 6$ mice per group, independent biological replicates; D15: Control vs. TAK-242: $p = 0.02$; **C** Control: $n = 5$ mice, independent biological replicates, TAK-242: $n = 6$ mice, independent biological replicates; D15: Control vs. TAK-242: $p = 0.018$. **D** Representative gating scheme for identification of M1-like (CD86$^+$CD206$^-$) and M2-like (CD86$^-$CD206$^+$) macrophages in total lung macrophages (CD45$^+$Ly6G$^-$CD64$^+$F4/80$^+$), alveolar macrophages (CD45$^+$Ly6G$^-$CD64$^+$F4/80$^+$SiglecF$^+$), and interstitial and recruited macrophages (CD45$^+$Ly6G$^-$CD64$^+$F4/80$^+$SiglecF$^-$) at day 20 after influenza infection in $EC^{Sparcl1-OE}$ mice treated with or without TAK-242, Macrophage gating strategies as shown in Supplementary Fig. 11A. **E** and **F** Quantification of the proportion of **E** M1-like (CD86$^+$CD206$^-$) and **F** M2-like (CD86$^-$CD206$^+$) macrophages in total lung macrophages (CD45$^+$Ly6G$^-$CD64$^+$F4/80$^+$), alveolar macrophages (CD45$^+$Ly6G$^-$CD64$^+$F4/80$^+$SiglecF$^+$) and interstitial and recruited macrophages (CD45$^+$Ly6G$^-$CD64$^+$F4/80$^+$SiglecF$^-$) at day 20 after influenza infection in WT and $EC^{Sparcl1-OE}$ mice treated with or without TAK-242, $n = 5$ mice per group, independent biological replicates. **E** $p = 0.04$ (total MΦ), $p = 0.038$ (alveolar MΦ). **F** $p = 0.045$ (interstitial + recruited MΦ). **G** The levels of pro-inflammatory cytokines, IL-1β, IL-6 and TNF in bronchoalveolar lavage fluid (BALF) were measured by ELISA in WT and $EC^{Sparcl1-OE}$ mice treated with or without TAK-242 at day 20 after influenza infection, $n = 5$ mice per group, independent biological replicates. IL-1β (Control: WT vs. $EC^{Sparcl1-OE}$: $p = 0.048$; $EC^{Sparcl1-OE}$ mice: Control vs. TAK-242: $p = 0.013$); IL-6 ($EC^{Sparcl1-OE}$ mice: Control vs. TAK-242: $p = 0.023$); TNF (WT mice: Control vs. TAK-242: $p = 0.011$; $EC^{Sparcl1-OE}$ mice: Control vs. TAK-242: $p = 0.033$). Data in **B**, **C**, **E**, and **F** are presented as means ± SEM, calculated using unpaired two-tailed $t$-test; Data in **G** are presented as means ± SEM, calculated using one-way analysis of variance (ANOVA), followed by Dunnett's multiple comparison test. *$P < 0.05$. Source data are provided as a Source Data file.

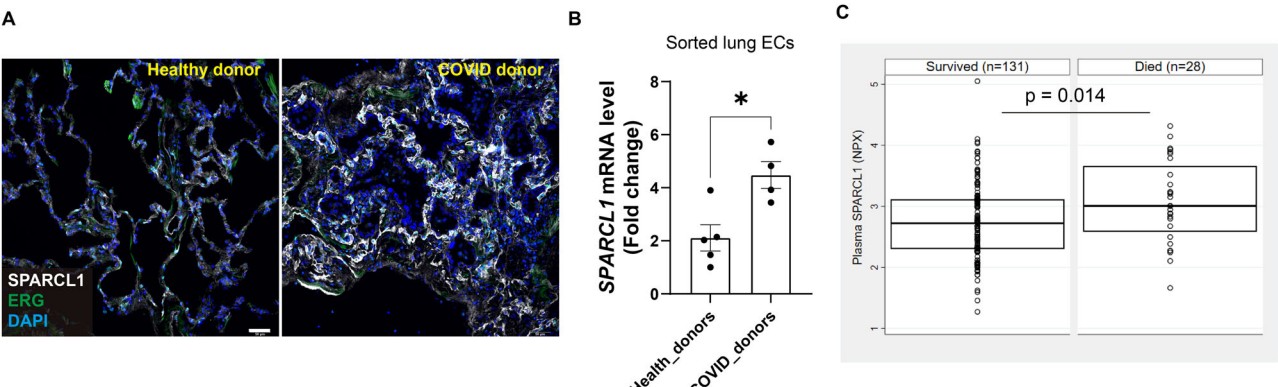

**Fig. 7 | High expression of SPARCL1 in COVID-19 patients correlates with increased mortality. A** Representative immunostaining image of SPARCL1 in endothelial cells (ERG) in both healthy and COVID-19 donors' lung tissue. Scale bar, 50 μm. **B** qPCR analysis of *SPARCL1* in isolated lung ECs (CD45$^-$EpCAM$^-$CD31$^+$) sorted from both healthy ($n = 5$) and COVID-19 ($n = 4$) donors' lung tissue, $p = 0.013$. **C** The SPARCL1 level in plasma from COVID-19 patients was measured by Olink proximal extension assay. For each group, the box depicts the median (center line) and upper and lower quartile. Upper quartile = 75th percentile and lower quartile = 25th percentile of the data. Data are presented as means ± SEM, Data in **B** were calculated using unpaired two-tailed $t$-test; Data in **C** were calculated using 2-sided, Wilcoxon Rank sum test comparing 90-day survivors ($n = 131$) to 90-day non-survivors ($n = 28$), *$P < 0.05$. Source data are provided as a Source Data file.

that assessing SPARCL1 levels in pneumonia patients serves not only as a biomarker of disease severity, but may allow for personalized medicine approaches. In patients with particularly high SPARCL1 levels, treatment with SPARCL1/TLR4/NF-κB antagonists could blunt uncontrolled inflammation, help them weather the cytokine storm, and ultimately promote better outcomes for these patients.

## Methods

### Patient samples
Human lung samples (both normal lung and COVID-19 samples) were obtained from Penn Lung Biology Institute Human Lung Tissue Bank (https://www.med.upenn.edu/lbi/htlb.html) as approved by the University of Pennsylvania Institutional Review Board Protocol #813685. COVID-19 tissue samples were from patients who previously tested positive for COVID-19 by PCR but tested negative via PCR multiple times prior to tissue acquisition. All COVID-19 samples were obtained from ventilated ARDS patients at least 30 days post-hospitalization who underwent lung transplant, at which time tissue samples were acquired.

Human plasma samples were from a prospective cohort study of participants with high risk for sepsis (Table S2) as approved by the University of Pennsylvania Institutional Review Board Protocol #808542. All human participants or their proxies provided written informed consent to participate. To be eligible, participants were admitted to the hospital and tested positive for SARS-CoV-2 by PCR test of nasal or respiratory secretions, as we have published[56,57].

Participants were excluded if they had been previously enrolled to the cohort, if they were chronically critically ill and residing in a long-term advanced-care hospital, if they desired exclusively palliative care on admission, or if the participant or their proxy were unable or unwilling to consent to the study. Blood was sampled within 3 days of hospital admission, processed immediately for plasma, and frozen at −80 °C until analysis. Plasma was not treated for viral inactivation prior to assay. Trained study personnel collected demographic and clinical data from the electronic health record (EHR) into case report forms. Participants were characterized by the World Health Organization ordinal scale for respiratory failure[58] at the time of blood sampling and considered severe respiratory failure if they required high flow oxygen (>6 lpm), non-invasive ventilation, or invasive ventilation (WHO ordinal scale ≥ 6) and moderate respiratory injury if they required no oxygen or oxygen at flow rates at or below 6 lpm (WHO ordinal scale ≤ 5)[56]. Mortality was assessed at 90 days using the EHR, which included a surveillance program post-discharge for patients discharged after COVID-19.

We assayed plasma proteins with the Olink proximal extension assay as described[56] and filtered results for SPARCL1. Protein concentrations were compared between categorical groups using the Wilcoxon rank sum test.

### Generation of *Sparcl1^flox^* mice
*Sparcl1^flox^* mice were generated using CRISPR-Cas9 strategy in mouse embryonic stem cells (ESCs). Briefly, a DNA template containing loxP

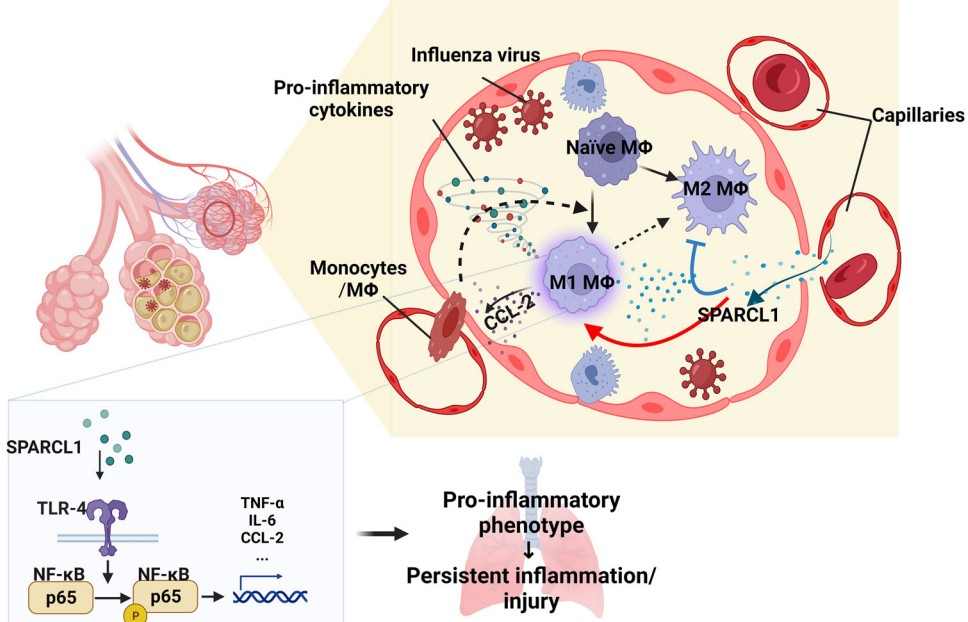

**Fig. 8 | Illustration of the impact of SPARCL1 on the progression of influenza pneumonia.** In influenza pneumonia, SPARCL1 from endothelial cells enters the alveoli via damaged blood vessels, prompting alveolar macrophages to adopt a pro-inflammatory (M1-like) state. This transformation hinges on SPARCL1 inducing TLR-4/NF-κB signaling activation in alveolar macrophages, leading to increased expression of pro-inflammatory cytokines. Simultaneously, heightened CCL-2 attracts more monocytes and macrophages into the alveoli. These recruited macrophages, once again exposed to SPARCL1 within the alveoli, further adopt a pro-inflammatory phenotype. The cumulative effect intensifies the local inflammatory response, causing tissue damage and hindering lung repair. Schematics and icons created with BioRender.com.

sites flanking exon 2 of the Sparcl1 "loxP-Sparcl1_Ex2-loxP" was synthesized by Genscript (Genscript Biotech Corp.) and purified. The gRNAs targeting the flanks of Sparcl1 exon2 were cloned into a CRISPR-Cas9 vector, pDG459 (Addgene, #100901), according to the protocol described previously with minor modifications[59]. Mouse ESCs(129/B6 F1 hybrid ES cell line V6.5)[60] were electroporated with linearized donor DNA (loxP-Sparcl1_Ex2-loxP) and pDG-459 with 2 gRNAs inserted, and clones were selected by puromycin (2 μg/ml) for 4–5 days. Following successful homologous recombination between the targeting donor DNA and ESC cell DNA as judged by PCR screening, targeted ESCs were then injected into C57BL/6J blastocysts to obtain chimeric mice following standard procedures. Chimeric mice (*Sparcl1*^flox/+^) are bred with *VECad*^CreERT2^ (*Cdh5*^CreERT2^) female mice[61] for direct characterization. Genotyping primers and PCR program are listed in Table S1.

### Generation of Sparcl1^OE^ (R26-LSL-Sparcl1) mice

Mouse Sparcl1 cDNA was amplified by PCR using the following primers: forward primers with 5' arm Xho I restrict enzyme cutting site added, mSparcl1-XhoI-F: ccgctcgagcggatgaaggctgtgtgctt ctcctc, reverse primers with 5' arm Sac II restrict enzyme cutting site added, mSparc1-SacII-R: tccccgcggggatcaaaagaggaggttttcatctat. Sparcl1 PCR products were cut, purified, and inserted into a generic targeting vector (pBigT, Addgene, #21270), pBigT-mSparcl1 was then subsequently cloned into a plasmid with the ROSA26 genomic flanking arms (ROSA26-PA, Addgene, #21271) following the protocol described previously[62], generating the final targeting vector ROSA-26-pBigT-mSparcl1 for homologous recombination. The linearized targeting vector was electroporated into mouse ESCs (same line as described above), and G418-resistant colonies were analyzed for proper editing by PCR. Finally, the targeted ESCs were injected into C57BL/6J blastocysts, and the resulting chimeric mice were bred with *VECad*^CreERT2^ (*Cdh5*^CreERT2^) female mice for direct characterization. Genotyping primers and PCR program are listed in Table S1.

### Animal treatments

C57BL6/J mice were bred in our own colony and were originally derived from Jackson Labs Stock #000664. All mice were *Sparcl1*^flox^ (noted as *Sparcl1*^flox/flox^) or *R26-LSL-Sparcl1* (noted as *Sparcl1*^+/WT^ or *Sparcl1*^+/+^) mice were crossed with *VECad*^CreERT2^ (*Cdh5*^CreERT2^) mice[61] to produce *VECad*^CreERT2^; *Sparcl1*^flox/flox^ mice, *VECad*^CreERT2^; *Sparcl1*^+/WT^, and *VECad*^CreERT2^; *Sparcl1*^+/+^ mice. *Sparcl1*^flox/flox^ mice lacking Cre or *VECad*^CreERT2^; *Sparcl1*^WT/WT^ mice were used as the conditional endothelial Sparcl1 knockout or overexpression control mice, respectively. *Tlr4*^−/−^ mice (Jax. Stock#029015) were kindly gifted by Dr. Igor E. Brodsky and Dr. Sunny Shin. Control animals were co-housed with experimental groups. These mice were administered five doses of tamoxifen (0.25 mg/g body weight) in 50 μl of corn oil every other day and rested for 2 weeks after the last injection, resulting in EC-specific deletion (*EC*^SparclI-KO^) or overexpression (*EC*^SparclI-OE^) of Sparcl1 in adult mice. Afterward, influenza virus A/H1N1/PR/8 was administered intranasally at 50–75 TCID50 units to mice according to experimental requirements as previously described[9,63]. Mice were weighed regularly and euthanized at the indicated time points for tissue harvest. In this study, all mice were used at 6–8 weeks old, and mice of both sexes were used in equal proportions, and both control and experimental animals were from the same parents and co-housed in a specific pathogen-free colony, the number of mice used for individual in vivo experiments is specified in the figure legends. All animals are euthanized using isoflurane in accordance with the University of Pennsylvania Animal Protocol (#806262). All animal experiments were carried out under the guidelines set by the University of Pennsylvania's Institutional Animal Care and Use Committees, protocol #806262, and followed all National Institutes of Health (NIH) Office of Laboratory Animal Welfare regulations.

### Cell culture conditions

Bone marrow-derived macrophages (BMDMs) were prepared as described previously[64]. Briefly, bone marrow cells were collected from

the femur and tibia of C57BL/6 or *Tlr4⁻/⁻* (TLR4-KO) mice and cultured in RPMI 1640 media with GlutaMAX Supplement added, containing 10% cosmic calf serum (CC; HyClone, #SH3008704), 1% penicillin/ streptomycin (P/S; Gibco, #15140122) and 20 ng/ml recombinant murine M-CSF (PeproTech, #315-02). THP-1 cells (ATCC TIB-202 ™) were kindly gifted by Dr. Mitchell Lab (Purchased from ATCC), cultured in RPMI 1640 media with GlutaMAX Supplement added, containing 10% cosmic calf serum, 1% penicillin/streptomycin (P/S), To induce differentiation into macrophages, THP-1 cells were treated with 100 ng/ml phorbol 12-myristate 13-acetate (PMA, STEMCELL Technologies, #74042) for 48 h. iMVECs were gifted by N. Mangalmurti. pLV-hSPARCL1 (VectorBuilder, #VB220607-1318rtk) was used with the pMD2.G and psPAX2 plasmids to produce lentiviral particles for generation of the SPARCL1-OE stable iMVECs cell line. Stable cell lines were selected by puromycin resistance (2 µg/ml) for 7 days. All cells were cultured in endothelial growth media (Lonza, #CC-3202). 293FT cells (Thermo Fisher Scientific) were cultured in Dulbecco's modified Eagle's medium (DMEM) (Thermo Fisher Scientific, #11965118) containing 10% cosmic calf serum and 1% penicillin/streptomycin (P/S). Depending on the experiment, BMDMs were treated with recombinant mouse SPARCL1 (0–20 µg/ml, R&D system, #4547-SL; SinoBiological, #50544-M08H); recombinant IL-4 (20 ng/ml, PeproTech, #214-14), lipopolysaccharides (50 ng/ml; Sigma Aldrich, #L6529), Resatorvid (TAK-242) (10 µM; MedChemExpress, #HY-11109), LPS-EB Ultrapure (LPS from *E. coli* O111:B4, InvivoGen, #tlrl-3pelps) or vehicle control.

## BALF cell count and total protein quantification
Bronchoalveolar lavage fluid (BALF) was collected by inserting a catheter in the trachea of euthanized mice. Lungs were infused with 1 ml PBS and gently retracted to maximize BALF retrieval and minimize shear forces. The fluid was centrifuged for 5 min at 500×*g*, the supernatant was collected for downstream experiments such as total protein quantification and ELISA, cell pellets were re-suspended, and red blood cells were removed using Red Blood Cell Lysis Buffer (Thermo Fisher Scientific, A1049201). The total cell count was determined using a cell counting chamber under light microscopy. Total protein in BALF was determined by the bicinchoninic acid (BCA) colorimetric assay using Pierce BCA Protein Assay Kit (Thermo Fisher Scientific, #23227).

## ELISA
BALF was collected in 1 ml PBS, as mentioned above. SPARCL1 (Mybiosource, #MBS2533471), TNF (Invitrogen, #88-7324-22), IL-6 (Invitrogen, #88-7064-22), IL-1β (Invitrogen, #88-7013-22), MCP-1/CCL-2 (Biolegend, # 432704), IFN-γ (Biolegend, #430807), IL-4 (Biolegend, #431107) and IL-10 (Biolegend, #431417) levels in mouse BALF were measured by ELISA assessments according to the manufacturer's instructions.

## Cytokine array
BALF was collected in 1 ml PBS, as mentioned above. Tested BALF was pooled from 4 mice of each group. The expression of cytokines was analyzed by the Proteome Profiler Mouse Cytokine Array Panel A Kit (ARY006, R&D Systems) according to the manufacturer's recommendation. Array panels were visualized using the ChemiDoc MP System (Bio-Rad). Protein levels were analyzed densitometrically using an image analysis system (Image J), corrected with values determined on positive controls and expressed as foldchange over the corresponding WT group.

## Whole lung cell suspension preparation
Human lung single-cell suspensions were prepared as described previously with slight modifications[65]. Briefly, distal lung tissue was obtained and dissected into roughly 5 cm³ pieces. Tissue was washed in 200 ml sterile PBS for 5 min at 4 °C at least two times or until PBS no longer appeared obviously bloody. An additional 5 min wash was then

performed with Hank's buffered saline solution (HBSS). Using autoclaved Kim Wipes, tissue was compressed to remove as much liquid as possible and further dissected into <1 cm³ pieces. Sterile HBSS buffer containing 5 U/ml Dispase II and 0.1 mg/ml DNase I + penicillin/streptomycin was added to the small tissue pieces. Tissue rapidly takes up the digest solution at this point, becoming visibly engorged. Tissue was digested on a shaker at 220 rpm at 37 °C for 2 h. Tissue was then liquified in digestion solution using an Osterizer Blender as follows: (low setting for all) 5 s milkshake, 3 s smoothie, and 5 s milkshake. The suspension was poured through a glass funnel lined with sterile 4 × 4 gauze, applying some compression to recover as much of the solution as possible. The cell suspension was sequentially filtered through 100, 70, and 40 µm strainers (Thermo Fisher Scientific). Finally, red blood cells were removed using a Red Blood Cell Lysis Buffer (Thermo Fisher Scientific, A1049201).

Lungs were harvested from mice, and single-cell suspensions were prepared as previously described[9]. Briefly, the lungs were thoroughly perfused with cold PBS via the left atrium to remove residual blood in the vasculature. Lung lobes were separated, collected, and digested with collagenase II (5 mg/ml in HBSS) (Worthington Biochemical, #LS004176) for 1 hour at 37 °C on the shaker at a speed of 200 rpm and mechanically dissociated by pipetting in sort buffer (DMEM + 2% CC + 1% P/S; referred to as "SB"). Next, cell suspensions were filtered by the 70-µm cell and treated with red blood cell lysis buffer containing 1:500 deoxyribonuclease I (DNase I) (MilliporeSigma, #D4527) for 5 min at room temperature, and the cell suspension was then used for subsequent experiments.

## Fluorescence-activated cell sorting and analysis
Whole lung single-cell suspensions were prepared as above; BMDMs were treated according to the experimental requirements, and Accutase (Sigma Aldrich, #A6964) was used to digest into single-cell suspension. Single-cell suspensions were then blocked in SB containing 1:50 human or mouse TruStain FcX™ for 5–10 min at 37 °C. The cell suspension was stained using cell viability dye (1:1000 in PBS, eBioscience™, #65-0865-18, intracellular FACS analysis use only) allophycocyanin (APC)/Cyanine7 or Brilliant Violet 421™ anti-human CD45 antibody (1:200, Biolegend, HI30), APC anti-human CD31 antibody (1:200, Biolegend, WM59) and PE anti-human CD326 (EpCAM) antibody (1:200, Biolegend, 9C4) for human lungs; Brilliant Violet 785 anti-mouse CD45 antibody (1:200; Biolegend, 30-F11), BD Horizon BUV395 Rat Anti-Mouse CD45(1:200; BD Bioscience, 30-F11), BD Horizon BUV563 Rat Anti-Mouse Ly-6G (1:200; BD Bioscience, 1A8), Brilliant Violet 605 anti-mouse Ly-6C antibody (1:100; Biolegend, HK1.4), FITC anti-mouse CD11c antibody (1:100; Biolegend, N418), PE/Cyanine5 anti-mouse I-A/I-E antibody (1:1000; Biolegend, M5/114.15.2), Brilliant Violet 785 anti-mouse CD274 (B7-H1, PD-L1) antibody (1:200; Biolegend, 10F.9G2), Brilliant Violet 711 anti-mouse/human CD11b antibody (1:200; Biolegend, M1/70), Brilliant Violet 421 anti-mouse F4/80 antibody(1:100; Biolegend, BM8), Alexa Fluor® 647 anti-mouse Siglec-F antibody (1:100; BD Bioscience, E50-2440), PE anti-mouse Ly-6G antibody(1:200; Biolegend, 1A8), PE/Cyanine7 anti-mouse CD64 (FcγRI) antibody (1:200, Biolegend, X54-5/7.1), Alexa Fluor 488 or PE–conjugated rat anti-mouse CD31 [platelet endothelial cell adhesion molecule 1 (PECAM1)] antibody (1:200; BioLegend, MEC13.3), Alexa Fluor 647 or FITC-conjugated rat anti-mouse CD326 (Ep-CAM) antibody (1:200; Biolegend, G8.8), PE/Cyanine5 anti-mouse CD86 antibody(1:200; Biolegend, GL-1), PE/Cyanine5 anti-mouse CD3ε antibody (1:200; Biolegend, 145-2C11), Alexa Fluor 700 anti-mouse/human CD45R/B220 antibody (1:50; Biolegend, RA3-6B2), BUV395 anti-mouse CD11b (1:200; BD Biosciences, M1/70), Brilliant Violet 421™ anti-mouse NK-1.1 antibody (1:100; Biolegend, PK136), PE-Cyanine7 CD127 Monoclonal antibody (1:100; Invitrogen, A7R34) for 45 min at 4 °C. Stained cells and "fluorescence minus one" (FMO) controls were then resuspended in SB + 1:1000 DNase + 1:1000 Draq7 (BioLegend, #424001) as

a live/dead stain. For measurement of apoptosis, BMDMs are dissociated into single-cell suspension by cell scraping and washed with cold PBS. Annexin V/7AAD staining was performed using APC Annexin V Apoptosis Detection Kit with 7-AAD (Biolegend, #640930) according to the manufacturers' directions. All flow analyses were performed on a BD FACSymphony A3 Cell Analyzer (BD Biosciences), and FACS sorting was performed on a BD FACSAria Fusion Sorter (BD Biosciences).

### Intracellular FACS analysis

Cell surface antigen antibodies were prepared and stained as described above, and then single-cell suspensions were fixed and permeabilized using Cyto-Fast™ Fix/Perm Buffer Set (Biolegend, #426803) according to the manufacturer's instructions. The cells were then stained with Alexa Fluor® 700 anti-mouse CD206 (MMR) antibody (1:200; Biolegend, C068C2) or RELM alpha monoclonal antibody (DS8RELM), Alexa Fluor™ 700 (1:200; Invitrogen, #56-5441-82), PE anti-mouse CXCL9 (MIG) antibody (1:100; Biolegend, #515603)for 30 min at room temperature; For intracellular EdU cytometry flow, mice were injected intraperitoneally with EdU (50 mg/kg; Santa Cruz Biotechnology, #sc-284628) at indicated time points. After euthanasia, the whole lung single-cell suspension was prepared and stained as above and fixed by 3.2% paraformaldehyde (PFA) (Electron Microscopy Sciences, #15714-S) for 15 min, washed twice using 3% bovine serum albumin (BSA) (in PBS), and permeabilized using 0.1% Triton X-100 (in PBS) for 15 min. EdU was detected using the Click-iT reaction coupled to an Alexa Fluor 647 azide following the instructions of the manufacturer (Invitrogen, #C10086). Intracellular flow analyses were performed on BD FACSymphony A3 Cell Analyzer (BD Biosciences).

### Immunofluorescence

For cryostat tissue sections, human lungs were obtained and transported to the laboratory on ice and mouse lungs were isolated and processed as previously described[63]. Freshly dissected lungs were fixed, embedded and cut into 7 μm-thick cryosections and postfixed for another 5 min with 3.2% PFA. Tissue sections were blocked in blocking buffer (1% BSA, 5% donkey serum, 0.1% Triton X-100, and 0.02% sodium azide in PBS) for 1 h at room temperature. For immunostaining of in vitro experiments, cells were cultured on 24-well chamber slides. At the experimental endpoint, the cells were fixed with 3.2% PFA for 15 min. EdU incorporation was analyzed using the Click-iT reaction coupled with an Alexa Fluor azide following the instructions of the manufacturer (Invitrogen, #C10086) and followed by subsequent immunostaining. Afterward, slides were probed with primary antibodies (CD31 1:200, Biolegend, MEC13.3; mSPARCL1 1:500, R&D systems, #AF2836-SP; hSPARCL1 1:500, R&D systems, #AF2728-SP; ERG 1:2000, Abcam, #ab92513; F4/80 1:200, Cell Signaling Technology, #30325; RELMα 1:200; Invitrogen, #56-5441-82; α-SMA 1:1000, Abcam, # ab32575) and incubated overnight at 4 °C. The next day, slides were washed and incubated with the fluorophore-conjugated secondary antibodies (typically Alexa Fluor conjugates, Life Sciences) at a 1:1000 dilution for ≥1 h. Last, slides were again washed, incubated with 1 μM 4′,6-diamidino-2-phenylindole (DAPI) for 5 min, and mounted using ProLong Gold (Life Sciences, #P36930). 4–6 images were taken randomly from each sample/section with a Leica Dmi8 microscope and analyzed with LAS X software (Leica).

### Histological analysis

Lung tissue sections fixed with 3.2% PFA were stained with Hematoxylin and Eosin Stain Kit (Vector Laboratories, #H-3502) according to the manufacturer's instruction and then imaged with a Leica DMi8 microscope. The quantification of H&E sections was performed under ×4 objective using LAS X tile scan mode and quantified by a previously described unbiased computational imaging approach[37]. Briefly, image pixel clusters were categorized according to staining intensity, and the tissue area was delineated into three distinct injury zones—namely, "severe," "damaged," and "normal"—for a robust and straightforward analysis. The established lung damage assessment program (https://github.com/WALIII/LungDamage) in MATLAB was then employed to quantify the portions of each zone.

### Pulse oximetry

Repeated measurements of peripheral oxygen saturation (SpO$_2$) were taken using a MouseOx Plus rat & mouse pulse oximeter and a MouseOx small collar sensor (Starr Life Sciences Corp.). Mice were shaved around the neck and shoulders where the collar sensor sits. Recordings were taken using MouseOx Premium Software (Starr Life Sciences Corp., Oakmont, PA, USA). Measurements were taken continuously for >3 min at a measurement rate of 15 Hz. Measurements were imported into Microsoft Excel, and all readings with a nonzero error code were filtered out. The average of these error-free readings was used to calculate the SpO$_2$ reading for each mouse for each given time point.

### Migration assay

Wound healing or migration assay was performed by seeding $2 \times 10^5$ cells into the Culture-Insert 2 well according to the manufacturer's recommendations (ibidi, #80209). After cells were attached, the insert was removed, and a low serum (<1%) medium with or without SPARCL1 protein was added. Images were taken immediately after insert removal (0 h) and after 24 h for the endpoint. Migration distance (in micrometers) was measured using LAS X software (Leica).

### Western blotting

Total protein from cells was extracted by lysis in radio-immunoprecipitation assay buffer (Santa Cruz Biotechnology, #sc-24948) with protease inhibitor cocktail (Cell Signaling Technology, #5872). Protein concentrations were determined using a BCA protein assay kit (Thermo Fisher Scientific, #23227). Samples with equal amounts of protein were fractionated on SDS–polyacrylamide gels (Bio-Rad, #4561084), transferred to polyvinylidene difluoride (Millipore Sigma, #IPVH00005) membranes, and blocked in 5% skim milk (Cell Signaling Technology, #9999s) in TBST (0.1% Tween 20 in tris-buffered saline) for 1.5 h at room temperature. The membranes were then incubated at 4 °C overnight with primary antibodies [phospho-NF-κB p65 1:1000, Cell Signaling Technology, #3033; NF-κB p65 1:1000, Cell Signaling Technology, #8242; mSPARCL1 1:500, R&D systems, #AF2836-SP; β-actin 1:2000, Cell Signaling Technology, #4970]. After the membranes were washed with TBST, incubations with 1:4000 dilutions (v/v) of the secondary antibodies were conducted for 2 h at room temperature. Protein expression was detected using the ChemiDoc XRS+ System (Bio-Rad). β-Actin was used as a loading control.

### RNA isolation and qPCR

Total RNA was extracted by ReliaPrep RNA Miniprep kit according to the manufacturer's recommendation (Promega, #Z6011(cell), #Z6111(tissue)) and then reverse-transcribed into complementary DNA using the iScript Reverse Transcription Supermix (Bio-Rad, #1708841). qPCR was performed using a PowerUp SYBR Green Master Mix and standard protocols on an Applied Biosystems QuantStudio 6 Real-Time PCR System (Thermo Fisher Scientific). Glyceraldehyde-3-phosphate dehydrogenase (GAPDH) was used to normalize RNA isolated from human cells (ECs and THP-1); RPL19 was used to normalize RNA isolated from mouse samples. The $2^{-\Delta\Delta Ct}$ comparative method was used to analyze expression levels. The primers used are listed in Table S1. For sorted mouse lung ECs, RNA was extracted and amplified (10 ng RNA/sample) using SMART-Seq® HT kit (Takara Bio, #634455) according to the manufacturer's instruction; the amplified cDNA was subsequently used for qPCR.

## Single-cell transcriptomics analysis

Influenza virus A/H1N1/PR/8 was administered intranasally to 6–8 week-old C57BL/6 mice at 50–60 U of median tissue culture infectious dose to mice (20–25 g, 50 U; 25–30 g, 60 U) as our previously described[9]. The whole lung single-cell suspension from mice on D0 ($n = 1$ mouse, male), D20 ($n = 1$ mouse, male) and D30 ($n = 1$ mouse, male) post-influenza infection were prepared as above, and FACS-sorted ECs were used for sequencing. Single-cell sequencing was performed on a 10X Chromium instrument (10X Genomics) at the Children's Hospital of Philadelphia Center for Applied Genomics. Cellranger mkfastq was used to generate demultiplexed FASTQ files from the raw sequencing data. Next, Cellranger count was used to align sequencing reads to the mouse reference genome (GRCm38) and generate single-cell gene barcode matrices. Post- processing and secondary analysis were performed using the Seurat package (v.4.0). First, variable features across single cells in the dataset were identified by mean expression and dispersion. Identifying variable features was then used to perform a PCA. The dimensionally reduced data was used to cluster cells and visualize using a UMAP plot. *Sparcl1* expression level was compared using the VlnPlot package in Seurat. Pseudotime analysis was conducted using Monocle3.

## Bulk RNA-seq analysis

Whole lung single-cell suspensions from $EC^{Sparcl1-OE}$ and WT mice on D20 post-influenza infection were prepared as above, and FACS-sorted macrophages (CD45$^+$/Ly6G$^−$/CD64$^+$/F4/80$^+$) were used for sequencing. RNA was extracted and amplified (10 ng RNA/sample) using SMART-Seq® HT kit (Takara Bio, #634455) according to the manufacturer's instruction, the amplified cDNA was quality checked, DNA libraries was prepared, and sequencing was performed on an Illumina HiSeq platform by GENEWIZ Co. Ltd. Raw data (raw reads) in fastq.gz format were processed through a general pipeline as describe previously[66]. Reads were aligned to the mm10 mouse genome using Kallisto and imported into R Studio for analysis via the TxImport package. Data was then normalized using the trimmed mean of M values normalization method in the EdgeR package. Mean-variance trend fitting, linear modeling, and Bayesian statistics for differential gene expression analysis were performed using the Voom, LmFit, and eBayes functions, respectively, of the Limma package, yielding differentially expressed genes between WT and $EC^{Sparcl1-OE}$ groups. Based on PCA analyses, a single WT sample was a clear outlier from all other samples in both groups, likely due to poor sort purity, and was removed from subsequent analysis. Volcano plots were created using the OmicStudio tools at https://www.omicstudio.cn/tool. All detectable genes derived from RNA-seq were used for gene set enrichment analysis (GSEA) using the Molecular Signatures Database (MSigDB) C2: curated gene sets according to the standard GSEA user guide (http://www.broadinstitute.org/gsea/doc/GSEAUserGuideFrame.html).

## Statistics

All statistical calculations were performed using GraphPad Prism 9. All in vitro experiments were repeated at least three times unless otherwise stated. Unpaired two-tailed Student's *t*-tests were used to ascertain statistical significance between the two groups. One-way analysis of variance (ANOVA) was used to assess statistical significance between three or more groups with one experimental parameter. For details on statistical analyses, tests used, size of *n*, definition of significance, and summaries of statistical outputs, see the corresponding figure legend and the "Results" section.

## Reporting summary

Further information on research design is available in the Nature Portfolio Reporting Summary linked to this article.

## Data availability

All data are included in the Supplementary Information or available from the authors, as are unique reagents used in this Article. The raw numbers for charts and graphs are available in the Source Data file whenever possible. The single-cell RNA sequencing data have been deposited in GEO under the accession code GSE201631. The Bulk RNA-seq data have been deposited in GEO under the accession code GSE225439. Source data are provided with this paper.

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

## Acknowledgements
We are grateful to the patients and families who agreed to participate and to the hospital staff who cared for them and facilitated our cohort study. We thank Dr. Igor E. Brodsky for generously providing the *Tlr4*$^{-/-}$ mice, and we appreciate the assistance provided by Dr. Wu Ling during the process of transferring the mice. We thank the Penn Lung Biology Institute Human Lung Tissue Bank, CHOP Flow Cytometry Core, and The Penn Vet Imaging Core (PVIC) for their assistance in performing these studies. We thank Dr. Noam Cohen for facilitating additional access to flow sorters. Finally, we thank BioRender for providing a platform to create the cartoons and schematics used in figures throughout this report. This work was supported by the following funding sources: NIH R01HL153539 (AEV), NIH R01HL164350 (AEV), NIH R01HL161196 (NJM), NIH R01HL155804 (NJM), R01HL155821 (EC).

## Author contributions
Conception and design: G.Z., N.J.M, M.J.M., A.E.V. Data acquisition: G.Z., L.X., A.I.W., M.E.G., C.V.C., S.A., J.W., X. L., S.K., N.P.H., M.C.B., K.M.S., J.D.P., E.C., J.D.C., M.M.C. Data analysis: G.Z., N.J.M, and A.E.V. Writing: G.Z., N.J.M., A.E.V. All authors read and approved the final manuscript.

## Competing interests
N.J.M. reports funding to her institution from NIH and Quantum Leap Healthcare Collaborative and serves on the scientific advisory board for Endpoint Health, Inc. The other authors declare no competing interests.
