## [Peer Review File · Nature Communications]

Editorial Note: Figure on page 16 of this Peer Review File have been redacted as indicated to remove third-party material where no permission to publish could be obtained.

REVIEWER COMMENTS

Reviewer #1 (Remarks to the Author):

In “Vascular Endothelial-derived SPARCL1 Exacerbates Viral Pneumonia Through Pro-Inflammatory Macrophage Activation,” the authors delineate a mechanism of vasculature involvement in immune responses to IAV infection in the lung. The authors first performed scRNA-seq on endothelial cells after IAV infection and detected upregulation of Sparcl1. Next, they implicate SPARCL1 in viral pneumonia severity through endothelial-specific Sparcl1 deletion and overexpression mouse models. Although the limits of subsetting macrophages as “M1” or “M2” are acknowledged, the authors go on to claim that SPARCL1 promotes M1 macrophage polarization. The authors then propose that these effects are induced via the TLR4 signaling cascade and show that in vivo blockade of TLR4 signaling using the inhibitor TAK-242 ameliorates the increased pneumonia in the in vivo SPARCL1 overexpression model. While the authors demonstrate an EC-specific phenotype that is subtle, but consistent, across model systems, many of the figures would be strengthened by including additional data (described in detail below). The most significant concern regarding the model presented in this manuscript is potential endotoxin contamination in the recombinant SPARCL1 protein used throughout the study. However, with additional positive and negative controls for in vitro experiments, this may be overcome as the in vivo data blocking TLR4 signaling convincingly demonstrates a role for TLR4 signaling in this model.

Overall, this study provides valuable insight into the role of a non-immune cell type in helping orchestrate immune responses in the lung. Endothelial cells are understudied in many contexts of infection and this work adds to our understanding of lung inflammation in response to viral injury. The manuscript should be revised with attention to the following comments:

Major concerns:

Figures 2-3: Most major cell types, except for neutrophils, are represented in the infiltrate panels in Figs 2, 3, & Supp Fig 3. Neutrophil data should be included in the manuscript for a more complete analysis of inflammation, especially given the increased infiltration of monocytes in Sparcl1-OE mice and the reliance throughout the manuscript on relative percentages of cell types (rather than cell numbers).

Figure 2: The H&E images in Fig 2L are not quantified nor is a systematic scoring system explained, yet in the text the authors state “These results indicate that mice lacking EC Sparcl1 exhibit dampened local inflammatory responses, further corroborated histologically, as these lungs bore less inflammatory immune cell infiltration and more orderly alveolar structure and thickness in the area of injury on day 25 post-infection.” To make these claims, the H&E images must be scored and quantified.

Figure 5: TLR4 activation in Fig 5 could be explained by endotoxin contamination in the purchased recombinant SPARCL1 protein preps. To help diminish this possible explanation, additional controls could be added for 5B, H, and I: (1) positive LPS controls to illustrate relative induction and (2) a recombinant protein from the same supplier that is not a TLR4 agonist and has the same level of potential endotoxin contamination as the rSPARCL1 used in that experiment. Furthermore, it is noteworthy that the authors use two different recombinant SPARCL1 proteins in their in vitro experiments, one that has <1EU of

endotoxin contamination, and one with <0.1EU. Because endotoxin contamination can be confounding when studying TLR4 signaling, the authors should denote which protein is used in each panel. Additionally, SPARCL1 dosage in 5A should be titrated down to show an increase in pho-P65 by western blot (rather than similar induction with all doses).

Figure 5K: The authors show that rSPARCL1 induces TLR4 signaling, but do not show data suggesting that SPARCL1 binds directly to TLR4. Therefore, the schematic can show SPARCL1 with an arrow to TLR4 but should not show direct SPARCL1 binding to TLR4.

Figures 2, 3, and 6: Although the authors quantify the M1/M2 polarization of macrophages in the ECSparcl1-OE in Supp Fig 5, they do not show relative AM and IM/recruited percentages at baseline. This should be quantified in both the ECSparcl1-OE and the ECSparcl1KO model. Additionally, the authors should show these same percentage metrics in Figure 6 for the percentages of CD45+ with and without TAK-242, in addition to the polarization metrics.

Minor concerns:

Throughout the paper, the authors use the terms Th1/Th2, including in the introduction and discussion. However, these terms relate to T cells which this manuscript does not explore. Perhaps the authors meant Type I and Type II responses? For example: "Instead of direct effects on the vasculature, our findings suggested that the high expression of SPARCL1 by ECs during pneumonia contributes to the worsening of lung injury by driving Th1 inflammation, ultimately causing more harm than benefit." As this sentence is not in reference to T cells, the terminology should be corrected.

Quotation marks are excessively and incorrectly used throughout the manuscript. For example, "cytokine storm" in quotes is acceptable, but "state", "angiostatic", "confluent", etc. should not be in quotation marks.

Additional labeling of EC subsets in Fig 1B-D and Supp Fig 1B is needed to provide clarity to the reader.

Figure 3: The survival curve for ECSparcl1-OE mice should also be included along with body weight and O₂.

Supp Fig 2: "especially" is qualitative and should not be used in a figure title.

In the discussion the authors state: "These results indicate that 1) macrophages are not permanently committed to a singular activation state, instead exhibiting the plasticity necessary to convert between polarized states and that 2) the activation status of macrophages, as well as the inflammatory microenvironment wherein they reside, are crucial factors influencing their response to and function within injured lungs." This sentence should be removed from the text because while both statements are known features of macrophages, they are not directly shown by this study.

In the final paragraph of the discussion, the language describing the nature of the findings should be softened a bit. While there is a consistent, demonstrable difference observed in morbidity in mice with

perturbations in EC Sparcl1 expression, it is still subtle and likely one of many contributors to pathology. For example, the terms “primarily” and “driver” should be changed or removed.

Reviewer #2 (Remarks to the Author):

The article by Zhao et al. presents the role of endothelial cell (EC)-derived SPARCL1 in altering lung macrophages to augment inflammation and lung injury during viral pneumonia. The significant findings of this study suggest that SPARCL1 is upregulated in gCap ECs, and this increase could contribute to SPARCL1 protein levels in BALF after influenza infection. Studies using EC-specific SPARCL1 knockout and overexpression mice suggest a pathogenic role for SPARCL1 that involves macrophage polarization defects during viral pneumonia. In particular, SPARCL1 induces lung macrophages towards the M1 phenotype via TLR4 binding and signaling. In support, in vivo studies using the TLR4 inhibitor TAK-242 show attenuation of SPARCL1-induced viral pneumonia. This study will have a significant impact as it describes the role of EC-derived SPARCL1 in macrophage polarization and viral pneumonia. Furthermore, limited data presented suggest high expression of SPARCL1 in endothelial cells of COVID-19 patient lungs and a possible association with increased mortality. Overall, the data presented in this manuscript points to a novel mechanism of regulating macrophage polarization by EC-derived SPARCL1 during viral infections. Importantly, all experiments were well done with appropriate experimental controls, but the author must address some critical experiments and questions.

Comments:

1. It is important to demonstrate the major source of SPARCL1 during influenza infection in the lungs. The data presented in both immunofluorescence staining and scRNA-seq data are not convincing enough to demonstrate ECs are the major source of SPARCL1. In Fig 1G and Supp. Fig.2, immunostaining for CD31 appears to show non-specific staining in all areas or cell types except airway epithelial cells, and it is important to include isotype controls. Importantly, co-staining of SPARCL1 with fibroblast-specific markers is needed to address SPARCL1 staining in peri-bronchial areas. Similarly, scRNA-seq data for SPARCL1 expression by other lung cells such as fibroblasts and immune cells are needed.
2. In Fig 2c, qPCR analysis shows the loss of SPARCL1 in EC; however, it is important to quantify the percentage of SPARCL1 loss in the total lung and BALF. Does the loss of SPARCL1 in EC have any effect on viral load in the lungs?
3. In Fig. 2k, the decrease in total MQ or interstitial/recruited MQ could be due to altered migration/invasion and/or proliferation caused by SPARCL1. It is important to assess the effects of SPARCL1 on the proliferation, apoptosis, and migration of MQ.
4. Does the overexpression of SPARCL1 in ECs alter macrophage subsets in the lung? If so, it is important to identify the mechanisms underlying SPARCL1-driven effects on macrophages.
5. It is important to use wildtype and SPARCL1 overexpression mice as control mice with no infection to compare the effects of SPARCL1 in the presence and absence of viral infection.

6. Is the observed M1-like macrophage polarization dependent or independent of Th1 or Th2 cytokines? Measuring additional Th1 and Th2 cytokines involved in macrophage polarization (such as IFN γ , IL-12, IL-10, IL-4, and IL-13) is necessary for studies using gain-of-function and loss-of-function mice for SPARCL1 with influenza infections. In vitro studies using ECs that are deficient or overexpressing SPARCL1 and co-cultured with macrophages are needed to assess the effects of SPARCL1 on macrophage polarization.

7. The effects of SPARCL1 on TLR4 signaling are intriguing, but caution is needed to reach such conclusions. Is the SPARCL1 used in these assays LPS-free? The use of polymyxin B is needed. Importantly, SPR binding studies are needed to demonstrate the binding stoichiometry and affinity between SPARCL1 and TLR4. Also, the use of TLR4-deficient macrophages is needed to assess the effects of SPARCL1 on M1 polarization.

8. Fig. 6 shows that TLR4 inhibition attenuates the pathological effects of SPARCL1 in EC-specific SPARCL1 overexpression mice. However, the potential non-specific effects of TAK-242 cannot be ruled out in these studies. Therefore, studies using TLR4-deficient macrophages in EC-specific SPARCL1 overexpression mice during influenza infection are needed.

9. Are the SPARCL1 levels elevated in the circulation of patients with influenza-induced pneumonia? The relevance of the data shown in Fig. 7 is not clear; it is important to use in vitro and in vivo infection models to establish the effects of SPARCL1 in COVID-19.

Reviewer #3 (Remarks to the Author):

What determines the severity of inflammation is an important question especially in the context of respiratory viral infections such as with Influenza or SARS-CoV-2 viruses. In the present study, Zhao et al report an interesting dataset showing that capillary endothelial cells are involved in the regulation of inflammation by the production of the secreted protein SPARCL1. They show that the production of this protein by those cells is associated with the worsening of the pneumonia symptoms. Mechanistically, they propose that SPARCL1 could induce a shift of macrophage polarization towards a pro-inflammatory phenotype. This signalling could proceed through TLR4 and pharmacological blockage of this pathway in mice impede the worsening of the symptoms upon infection of mice overexpressing SPARCL1 in VECad+ cells. Finally, they show that expression of SPARCL1 is increased in lungs of COVID-19 patients. Moreover, among infected patients, expression is higher in plasma from patients who died than in the ones who survived. Altogether, the work is nicely performed and the hypothesis is interesting and promising.

However, several major points need to be addressed in order to be fully convincing.

Major comments

- The authors present dev.ECs and Immu.ECs as two distinct subsets of cells. Moreover, upon infection, they report diminution of Dev.ECs and increase of Immu.ECs and the opposite (even if less clear) upon recovery. However, at steady state, Immu.ECs are nearly absent. One alternative hypothesis is that it could be the same cells that are activated by the infection and display an alternative transcription program.

A trajectory analysis of these single cell data should be performed in order to decipher the dynamics of these populations over time.

- One limitation of the study relates to the cells that are producing SPARCL1. Fig1 and S1 are not convincing enough to identify the cells that are producing the protein. Of course, the protein is at least produced by capillary ECs and VE-Cad expressing cells (as shown by the models afterwards), however, it would have been nice to use flow cytometry and intracellular staining for SPARCL1 in order to quantify the importance of ECs in SPARCL1 production at steady state and upon infection. Enlargement of the pictures presented in those figures would also be necessary. Finally, the regions that are enlarged should be shown on the lower magnification pictures.

- Related to this latter point, the authors should provide an explanation for the fact that, at the mRNA level, SPARCL1 is still produced a lot by ECs at 30 days post infection (and the two subtypes are still very well present). However, at the protein level, SPARCL1 production in BALF is very much reduced. Could it suggest that other cells are the main producers of SPARCL1? Accordingly, mice in which ECs do not or overexpress SPARCL1 display different phenotypes but they are not dramatically different suggesting that SPARCL1 could be produced by other cells that would drive most of the effect.

- The authors show differences at the transcriptomic level until d30 post-infection but do not test later time points. Does it come back to steady state levels? Does the Immu.ECs disappear? Do epigenetic marks persist in those cells?

- It seems that the gating strategies presented in Suppl Fig. 3 and 4 could be improved:

- o Some cells of the lung are very autofluorescent. Especially, alveolar macrophages. It therefore appears that the gate for live cells should be increased to the CD45+ APC-Cy7 dim cells that are likely autofluorescent cells. Alternatively, a FMO control for live/dead should be provided.

- o The study highlights an effect of SPARCL1 secretion by ECs of macrophages but do not identify monocytes at any point. This is a major limitation of the paper as those cells are very important upon viral infections (Maquet al. doi: 10.1126/sciimmunol.abn3240; Li et al. doi: 10.1126/sciimmunol.abj5761). Moreover, it identifies two populations of myeloid cells/ macrophages, namely alveolar macrophages and interstitial/recruited macrophages. Firstly, the gating strategy is not shown, however, this is central to the paper (some SiglecF stainings are shown later but a clear gating strategy with all the markers including SiglecF, F4/80, CD11c, CD11b and CD64 is needed). Secondly, this dichotomy is not precise enough. At least, AMs, IMs and monocytes should be distinguished. However, IMs are themselves composed of several subsets that could improve the message of the paper (See Schyns et al, doi: 10.1038/s41467-019-11843-0 or Chakarov et al. doi: 10.1126/science.aau0964). Besides, inflammatory DCs are also cells derived from monocytes that play an important role upon viral infection into the lung (Bosteels et al., doi: 10.1016/j.immuni.2020.04.005), identifying those cells would also be very important. Those different populations should be clearly distinguished and the related graphs should be modified accordingly.

- o In the Figure S5, why not taking CD11b+SiglecF+ cells that are recruited cells in differentiation to AMs?

In Figure S5, it seems that the gate on F4/80 FMO should be more on the left side as it discards a lot of cells of interest (recruited macrophages) as located

- o The authors identify and quantify ILCs which are important but rather small populations but do not quantify neutrophils that are directly related to the severity of viral infections. Here too, several subsets could be identified and the effect of SPARCL1 on those cells should be addressed.

- o Figures 4 and S5 compare the expression of CD86 and CD206 in different populations. This could be interesting. However, stainings are not convincing, especially, CD206 staining should be improved. The separation of the different subsets of cells based on CD206 appears as a weakness of the paper. But this is central for the actual distinction between “M1” and “M2” macrophages.

- The effect of SPARCL1 on other cells is very interesting. However, the sole dichotomy between M1 and M2 orientated cells is too simple. Ideally, a single cell experiment on myeloid cells at day 12 in WT, EC-Sparcl1-KO and EC-SPARKL1-OE would be very informative. Alternatively, showing expression of key regulatory markers (such as PDL1 or CXCL9 (Maquet al. doi: 10.1126/sciimmunol.abn3240) and others) by myeloid cells could improve a lot the message.

- The absence of SPARCL1 effect on populations of myeloid cells at steady state (Fig. S5B) is explained as a consequence of the fact that SPARCL1 fails to infiltrate the alveolar space in uninfected mice. This is not clear. Indeed, only AMs are in the alveolae. Moreover, could it not be that only recruited cells display plasticity and not resident ones at steady state? This should be at least discussed.

- The authors show that there is no difference in the percentages of different cells (T, B, ...) between the different groups they test (WT, EC-Sparcl1-KO and EC-SPARKL1-OE). However, they do not analyze if those cells exhibit different functional properties (regulatory vs cytotoxic for example). This should be at least discussed.

- The authors suggest that SPARCL1 acts through TLR4. If it has an effect on recruited monocyte-derived cells, the ideal model would be CCR2CreERT2 TLR4flox mice. This could be tested in order to be fully convincing. This could allow to specifically block TLR4 signalling (and therefore SPARCL1 effect if this is the pathway involved) at precise time points.

Minor comments

- In the introduction, the increased expression of SPARCL1 after influenza infection is presented as a fact but no reference is provided. Moreover, this result is presented in the result section. This is not clear and should therefore be modified.

- The initial single cell experiment should be described in a dedicated chapter in the material and methods. Indeed, at the moment, it appears complicated to know the number of mice that were used, their sex and the dose of virus with which they have been infected.

- In Figure 1, the EC subsets should be shown as indicated on Figure S1

- Spaces are missing before references at several locations

- The authors have made several mutant mice, which is rather impressive. For the EC-Sparcl1-KO, what is the PCR used in Figure 2C? Could the authors identify those primers on Figure 2A?

- The authors measure different cytokines that are interesting to measure inflammation but some difference could also relate in the recruitment of monocytes. Therefore, it seems that measurements of BALF and circulating CCL2 could also be interesting.

- Fig. 4D is not convincing. The difference in RELM α expression is very different but much bigger than the one reported on FACS with identification of M2 macrophages. Quantification of RELM α + cells by FACS should be performed.

- Fig. 5: in the text, the authors suggest differences of BMDM orientation based on morphology. This is not convincing.
- In the discussion, the authors talk about differences in Th1 inflammation as a fact, however, this is not tested.

Response to Reviewers

We have provided a point-by-point response plan below. I have kept the Reviewers' comments in blue and our response in normal black text.

REVIEWER COMMENTS

Reviewer #1 (Remarks to the Author):

In "Vascular Endothelial-derived SPARCL1 Exacerbates Viral Pneumonia Through Pro-Inflammatory Macrophage Activation," the authors delineate a mechanism of vasculature involvement in immune responses to IAV infection in the lung. The authors first performed scRNA-seq on endothelial cells after IAV infection and detected upregulation of Sparcl1. Next, they implicate SPARCL1 in viral pneumonia severity through endothelial-specific Sparcl1 deletion and overexpression mouse models. Although the limits of subsetting macrophages as "M1" or "M2" are acknowledged, the authors go on to claim that SPARCL1 promotes M1 macrophage polarization. The authors then propose that these effects are induced via the TLR4 signaling cascade and show that in vivo blockade of TLR4 signaling using the inhibitor TAK-242 ameliorates the increased pneumonia in the in vivo SPARCL1 overexpression model. While the authors demonstrate an EC-specific phenotype that is subtle, but consistent, across model systems, many of the figures would be strengthened by including additional data (described in detail below). The most significant concern regarding the model presented in this manuscript is potential endotoxin contamination in the recombinant SPARCL1 protein used throughout the study. However, with additional positive and negative controls for in vitro experiments, this may be overcome as the in vivo data blocking TLR4 signaling convincingly demonstrates a role for TLR4 signaling in this model.

Overall, this study provides valuable insight into the role of a non-immune cell type in helping orchestrate immune responses in the lung. Endothelial cells are understudied in many contexts of infection and this work adds to our understanding of lung inflammation in response to viral injury. The manuscript should be revised with attention to the following comments:

Thank you for the Reviewer's recognition of our work; we hope that our additional data will address remaining concerns raised by the Reviewer.

Major concerns:

Figures 2-3: Most major cell types, except for neutrophils, are represented in the infiltrate panels in Figs 2, 3, & Supp Fig 3. Neutrophil data should be included in the manuscript for a more complete analysis of inflammation, especially given the increased infiltration of monocytes in Sparcl1-OE mice and the reliance throughout the manuscript on relative percentages of cell types (rather than cell numbers).

We thank the Reviewer for this suggestion. We have now added the neutrophil data (**Suppl Fig.6B and 8A**). We did not observe any significant differences in neutrophils under any conditions.

Figure 2: The H&E images in Fig 2L are not quantified nor is a systematic scoring system explained, yet in the text the authors state “These results indicate that mice lacking EC Sparcl1 exhibit dampened local inflammatory responses, further corroborated histologically, as these lungs bore less inflammatory immune cell infiltration and more orderly alveolar structure and thickness in the area of injury on day 25 post-infection.” To make these claims, the H&E images must be scored and quantified.

We appreciate this suggestion. We have corrected the wording and employed a recently published (Liberti et al., 2021), unbiased computational quantitative method for evaluation of these pathological sections (**Fig.2M**). The detailed evaluation methods are provided in the Materials and Methods section.

Figure 5: TLR4 activation in Fig 5 could be explained by endotoxin contamination in the purchased recombinant SPARCL1 protein preps. To help diminish this possible explanation, additional controls could be added for 5B, H, and I: (1) positive LPS controls to illustrate relative induction and (2) a recombinant protein from the same supplier that is not a TLR4 agonist and has the same level of potential endotoxin contamination as the rSPARCL1 used in that experiment. Furthermore, it is noteworthy that the authors use two different recombinant SPARCL1 proteins in their in vitro experiments, one that has <1EU of endotoxin contamination, and one with <0.1EU. Because endotoxin contamination can be confounding when studying TLR4 signaling, the authors should denote which protein is used in each panel. Additionally, SPARCL1 dosage in 5A should be titered down to show an increase in pho-P65 by western blot (rather than similar induction with all doses).

We agree that it is critical to address the potential for LPS contamination. To exclude the possibility of LPS contamination in the recombinant SPARCL1 protein, we employed the LPS inhibitor, Polymyxin B, which significantly inhibited LPS-induced NF- κ B phosphorylation. However, even at relatively high concentrations (100 μ g/ml), it did not impede SPARCL1-induced NF- κ B phosphorylation (**Suppl Fig.14A-B**). This data rules out LPS contamination in SPARCL1 as an accidental activator of TLR4. In addition, we tested recombinant SPARCL1 proteins from two different brands at concentrations ranging from 0 to 5 μ g/ml for their impact on NF- κ B activation (**Suppl Fig.14A**). Our results indicate that both brands significantly induced phosphorylation of NF- κ B P65, with a noticeable effect at approximately 1 μ g/ml. Following the reviewer's suggestion, we have added the brand information for all recombinant protein experiments to the corresponding Figure legends. Finally, we used the conditioned medium from SPARCL1-overexpressing endothelial cells and successfully induced a pro-inflammatory shift in THP-1 macrophages (**Suppl Fig.14C-G**). The above evidence strongly confirms that SPARCL1 from endothelial cells can activate

the NF- κ B signaling pathway, thereby promoting macrophage polarization towards the M1 phenotype.

Additionally, as we noted and cited in our original submission, it is important to take into consideration that SPARCL1 has been shown to directly bind and activate TLR4 in biochemical assays in hepatocytes (Liu et al 2021, *JCI*, doi: 10.1172/JCI144801), providing substantial additional support to this mechanism. See our related response to Reviewer 2 below.

Figure 5K: The authors show that rSPARCL1 induces TLR4 signaling, but do not show data suggesting that SPARCL1 binds directly to TLR4. Therefore, the schematic can show SPARCL1 with an arrow to TLR4 but should not show direct SPARCL1 binding to TLR4.

Thank you for pointing that out. We have made corrections to the schematic diagram as suggested, though again note the above publication has indeed demonstrated binding of SPARCL1 to TLR4.

Figures 2, 3, and 6: Although the authors quantify the M1/M2 polarization of macrophages in the EC $Sparcl1$ -OE in Supp Fig 5, they do not show relative AM and IM/recruited percentages at baseline. This should be quantified in both the EC $Sparcl1$ -OE and the EC $Sparcl1$ KO model. Additionally, the authors should show these same percentage metrics in Figure 6 for the percentages of CD45+ with and without TAK-242, in addition to the polarization metrics.

We thank the reviewer's comments. We have provided all these data as suggested, please see **Suppl Fig.3J, 7H and 16C**.

Minor concerns:

Throughout the paper, the authors use the terms Th1/Th2, including in the introduction and discussion. However, these terms relate to T cells which this manuscript does not explore. Perhaps the authors meant Type I and Type II responses? For example: "Instead of direct effects on the vasculature, our findings suggested that the high expression of SPARCL1 by ECs during pneumonia contributes to the worsening of lung injury by driving Th1 inflammation, ultimately causing more harm than benefit." As this sentence is not in reference to T cells, the terminology should be corrected.

We thank the Reviewer for pointing that out. We have made the changes, removing, or replacing Th1/Th2-related statements.

Quotation marks are excessively and incorrectly used throughout the manuscript. For example, "cytokine storm" in quotes is acceptable, but "state", "angiostatic", "confluent", etc.

should not be in quotation marks.

Thanks for pointing this out, we have corrected this throughout the manuscript.

Additional labeling of EC subsets in Fig 1B-D and Supp Fig 1B is needed to provide clarity to the reader.

We have labeled all these EC subsets as suggested.

Figure 3: The survival curve for ECSparcl1-OE mice should also be included along with body weight and O2.

We have added this Survival curve data in **Fig.3F**.

Supp Fig 2: “especially” is qualitative and should not be used in a figure title.

We have corrected this.

In the discussion the authors state: “These results indicate that 1) macrophages are not permanently committed to a singular activation state, instead exhibiting the plasticity necessary to convert between polarized states and that 2) the activation status of macrophages, as well as the inflammatory microenvironment wherein they reside, are crucial factors influencing their response to and function within injured lungs.” This sentence should be removed from the text because while both statements are known features of macrophages, they are not directly shown by this study.

We agree with the reviewer’s comments and have deleted these sentences.

In the final paragraph of the discussion, the language describing the nature of the findings should be softened a bit. While there is a consistent, demonstrable difference observed in morbidity in mice with perturbations in EC Sparcl1 expression, it is still subtle and likely one of many contributors to pathology. For example, the terms “primarily” and “driver” should be changed or removed.

We have made all these changes as suggested.

Reviewer #2 (Remarks to the Author):

The article by Zhao et al. presents the role of endothelial cell (EC)-derived SPARCL1 in

altering lung macrophages to augment inflammation and lung injury during viral pneumonia. The significant findings of this study suggest that SPARCL1 is upregulated in gCap ECs, and this increase could contribute to SPARCL1 protein levels in BALF after influenza infection. Studies using EC-specific SPARCL1 knockout and overexpression mice suggest a pathogenic role for SPARCL1 that involves macrophage polarization defects during viral pneumonia. In particular, SPARCL1 induces lung macrophages towards the M1 phenotype via TLR4 binding and signaling. In support, in vivo studies using the TLR4 inhibitor TAK-242 show attenuation of SPARCL1-induced viral pneumonia. This study will have a significant impact as it describes the role of EC-derived SPARCL1 in macrophage polarization and viral pneumonia. Furthermore, limited data presented suggest high expression of SPARCL1 in endothelial cells of COVID-19 patient lungs and a possible association with increased mortality. Overall, the data presented in this manuscript points to a novel mechanism of regulating macrophage polarization by EC-derived SPARCL1 during viral infections. Importantly, all experiments were well done with appropriate experimental controls, but the author must address some critical experiments and questions.

Thank you for the Reviewer's recognition of our work; we hope that our additional data will address remaining concerns raised by the Reviewer.

Comments:

1. It is important to demonstrate the major source of SPARCL1 during influenza infection in the lungs. The data presented in both immunofluorescence staining and scRNA-seq data are not convincing enough to demonstrate ECs are the major source of SPARCL1. In Fig 1G and Supp. Fig.2, immunostaining for CD31 appears to show non-specific staining in all areas or cell types except airway epithelial cells, and it is important to include isotype controls. Importantly, co-staining of SPARCL1 with fibroblast-specific markers is needed to address SPARCL1 staining in peri-bronchial areas. Similarly, scRNA-seq data for SPARCL1 expression by other lung cells such as fibroblasts and immune cells are needed.

Thank you for the Reviewer's suggestions. We have added more data as suggested, including lung single-cell transcriptome data (**Suppl Fig.2A**) and co-staining with the stromal cell marker α -SMA and SPARCL1 (**Fig.2E**). Additionally, we performed intracellular flow cytometric analysis of SPARCL1 (**Suppl Fig.2G**). Our data indicate that SPARCL1 is primarily derived from endothelial cells and stromal cells. The loss of endothelial *Sparcl1* during infection directly results in changes in SPARCL1 concentration in BALF and in total lung tissue homogenate (**Suppl Fig.3D-E**), confirming that endothelial cells are a major, biologically relevant source of SPARCL1, especially in pathological conditions.

Regarding CD31 staining, as the lung is a highly vascularized organ, the alveoli are completely surrounded by dense microvasculature. The CD31 antibody used is a common and well-established antibody, and the staining in the figure represents a normal vascular pattern. To address the reviewer's concerns, we also included an isotype control, as shown below, confirming the specificity of our CD31 staining. (This data has not been included in the Supplementary Figure, but we are happy to include it if needed.)

2. In Fig 2c, qPCR analysis shows the loss of SPARCL1 in EC; however, it is important to quantify the percentage of SPARCL1 loss in the total lung and BALF. Does the loss of SPARCL1 in EC have any effect on viral load in the lungs?

Thank you for these suggestions. We have provided this data, including SPARCL1 levels in BALF and total lung before and during infection in WT and EC^{SPARCL1-KO} mice (**Suppl Fig.3D and E**). We also assessed viral loads in the lungs by detecting the expression of the influenza virus-specific gene M, observing no apparent changes dependent on SPARCL1 (**Suppl Fig.3K**).

3. In Fig. 2k, the decrease in total MQ or interstitial/recruited MQ could be due to altered migration/invasion and/or proliferation caused by SPARCL1. It is important to assess the effects of SPARCL1 on the proliferation, apoptosis, and migration of MQ.

These are excellent questions. We investigated the impact of recombinant SPARCL1 protein on the proliferation, migration, and apoptosis of BMDMs. We did not observe any obvious effects of SPARCL1 on macrophage proliferation, migration, or apoptosis. However, intriguingly, SPARCL1 demonstrated an appreciable protective effect against H₂O₂-induced apoptosis (**Suppl Fig.10**), which we hope to explore in future studies.

4. Does the overexpression of SPARCL1 in ECs alter macrophage subsets in the lung? If so, it is important to identify the mechanisms underlying SPARCL1-driven effects on macrophages.

We are a bit confused by this question as this study already provided extensive data focused on this question. Indeed, SPARCL1 overexpression does not impact homeostatic macrophage subsets (**Suppl. Fig 7H**) but has a significant impact on relative proportions of alveolar vs. interstitial / recruited macrophages and polarization of these macrophage type

upon influenza infection (**Fig. 3N and 4B-C**). Mechanistically, we focused on determining how SPARCL1 induces macrophages to adopt a pro-inflammatory phenotype through the TLR-4/NF- κ B pathway. We also assessed additional myeloid and immune cell populations, without noticing significant differential changes. We agree that there might be more subtle alterations in specific macrophage subtypes, such as monocyte-derived macrophages, since we noticed an increased/decreased CCL-2 level in BALF (**Fig.2K and 3M, Suppl Fig.17**) after EC Sparcl1 overexpression or loss during injury, but given the extensive experimentation we've performed, further analysis of more subtle phenotypic changes seems outside the scope of this work.

5. It is important to use wildtype and SPARCL1 overexpression mice as control mice with no infection to compare the effects of SPARCL1 in the presence and absence of viral infection.

We have provided this data in **Suppl Fig.7A-B**.

6. Is the observed M1-like macrophage polarization dependent or independent of Th1 or Th2 cytokines? Measuring additional Th1 and Th2 cytokines involved in macrophage polarization (such as IFN γ , IL-12, IL-10, IL-4, and IL-13) is necessary for studies using gain-of-function and loss-of-function mice for SPARCL1 with influenza infections. In vitro studies using ECs that are deficient or overexpressing SPARCL1 and co-cultured with macrophages are needed to assess the effects of SPARCL1 on macrophage polarization.

We conducted further testing for more cytokines in BALF as suggested (**Fig.2D-H, 3I-M**). Additionally, we performed additional cytokine arrays in both EC loss and overexpression of Sparcl1 during infection (**Suppl Fig.3F-H,7C-E**). Regarding the author's suggestion of co-culturing macrophages with ECs loss or overexpression of Sparcl1, we appreciate the reviewer's advice and agree that that is an excellent experiment. As SPARCL1 is a secreted protein, we constructed an endothelial cell line with *SPARCL1* overexpression (SPARCL1-OE iMVECs) and collected the conditioned medium for co-culture with THP-1 macrophages. Our data showed that the SPARCL1-OE conditional medium significantly induced the phosphorylation of NF-Kb p65 and expression of pro-inflammatory cytokines *TNFA* and *IL-6*, indicating that endothelial cell-derived SPARCL1 induces a pro-inflammatory shift in macrophages (**Suppl Fig.14C-G**), consistent with our *in vivo* data. In our hands wild type lung endothelial cells express very low levels (undetectable) of SPARCL1 (**Suppl Fig. 14D**), we believe that culturing macrophages with SPARCL1-KO endothelial cells would not be meaningful.

7. The effects of SPARCL1 on TLR4 signaling are intriguing, but caution is needed to reach such conclusions. Is the SPARCL1 used in these assays LPS-free? The use of polymyxin B is needed. Importantly, SPR binding studies are needed to demonstrate the binding stoichiometry and affinity between SPARCL1 and TLR4. Also, the use of TLR4-deficient macrophages is needed to assess the effects of SPARCL1 on M1 polarization.

We thank the Reviewer for these valuable suggestions and completely agree that it is important to assess the potential role of contaminating LPS. To exclude the possibility of

LPS contamination in the recombinant SPARCL1 protein, we employed the LPS inhibitor, Polymyxin B, which significantly inhibited LPS-induced NF- κ B phosphorylation. However, it did not appreciably impede SPARCL1-induced NF- κ B phosphorylation (**Suppl Fig.14A-B**). This data ruled out LPS contamination in SPARCL1. Regarding the reviewer's suggestion to use SPR experiments to confirm the direct binding of SPARCL1 with TLR-4, previous studies have already provided comprehensive data supporting the direct interaction of SPARCL1 with TLR4 (**see below**). Additionally, we conducted tests in both WT and TLR4 KO BMDMs, and observed that TLR4KO BMDMs show no response to either LPS or SPARCL1 in terms of NF- κ B activation. Considering these existing findings and our experimental results, additional biochemical assays to demonstrate direct binding seem somewhat redundant given the excellent work already performed in the Liu et al. manuscript.

[Figure Redacted]

← The immunoprecipitation (IP) experiments revealed a direct interaction/binding between SPARCL1 and TLR4 (Liu, Bin, et al. *The Journal of Clinical Investigation*. 2021).

8. Fig. 6 shows that TLR4 inhibition attenuates the pathological effects of SPARCL1 in EC-specific SPARCL1 overexpression mice. However, the potential non-specific effects of TAK-242 cannot be ruled out in these studies. Therefore, studies using TLR4-deficient macrophages in EC-specific SPARCL1 overexpression mice during influenza infection are needed.

We agree that to validate whether *in vivo* endothelial overexpression of *Sparcl1* induces macrophages to shift towards a pro-inflammatory state through TLR4, it would be necessary to utilize macrophage-specific knockout of TLR4 while simultaneously overexpressing SPARCL1 in endothelial cells. However, due to technical challenges associated with achieving cell-specific overexpression and knockout using the "Cre-loxP" system, and the lack of commercially available "CCR2-DreERT; TLR4 rox" or "CCR2-FlpOERT2; TLR4 frt" mice, we find this approach challenging. Utilizing only our existing mice would result in simultaneous TLR4 deletion and SPARCL1 overexpression in both endothelial cells and macrophages, making any experimental results uninterpretable. This issue was also highlighted by Reviewer 3. In light of these challenges, we have added this discussion to the "Limitations of the study" section.

9. Are the SPARCL1 levels elevated in the circulation of patients with influenza-induced pneumonia? The relevance of the data shown in Fig. 7 is not clear; it is important to use in vitro and in vivo infection models to establish the effects of SPARCL1 in COVID-19.

We appreciate the Reviewer's suggestion. However, obtaining serum samples from individual influenza-induced severe or fatal cases is very limited in clinical practice. Additionally, conducting *in vivo* and *in vitro* experiments on SARS-CoV-2 infection requires a specialized laboratory with Biosafety Level 3 (ABSL-3) containment due to its high pathogenicity and contagious nature. Our routine laboratory facilities are typically ABSL-1 or 2, lacking the conditions necessary for such experiments. We hope the reviewer understands these constraints, and we have attempted to be careful not to overstate our findings.

Reviewer #3 (Remarks to the Author):

What determines the severity of inflammation is an important question especially in the context of respiratory viral infections such as with Influenza or SARS-CoV-2 viruses. In the present study, Zhao et al report an interesting dataset showing that capillary endothelial cells are involved in the regulation of inflammation by the production of the secreted protein SPARCL1. They show that the production of this protein by those cells is associated with the worsening of the pneumonia symptoms. Mechanistically, they propose that SPARCL1 could induce a shift of macrophage polarization towards a pro-inflammatory phenotype. This signalling could proceed through TLR4 and pharmacological blockage of this pathway in mice impede the worsening of the symptoms upon infection of mice overexpressing SPARCL1 in VECad+ cells. Finally, they show that expression of SPARCL1 is increased in lungs of COVID-19 patients. Moreover, among infected patients, expression is higher in plasma from patients who died than in the ones who survived. Altogether, the work is nicely performed and the hypothesis is interesting and promising. However, several major points need to be addressed in order to be fully convincing.

We appreciate the Reviewer's positive recognition of our work. We hope our revisions provide a satisfactory response to the Reviewer's important concerns.

Major comments

- The authors present dev.ECs and Immu.ECs as two distinct subsets of cells. Moreover, upon infection, they report diminution of Dev.ECs and increase of Immu.ECs and the opposite (even if less clear) upon recovery. However, at steady state, Immu.ECs are nearly absent. One alternative hypothesis is that it could be the same cells that are activated by the infection and display an alternative transcription program. A trajectory analysis of these single cell data should be performed in order to decipher the dynamics of these populations over time.

We actually completely agree that Dev.ECs and Immu.ECs are essentially the same cell type in two different activation states, and we attempted to indicate that in the Discussion. As suggested, we provided the pseudotime analysis of these 2 gCap ECs in **Suppl Fig.1E**.

- One limitation of the study relates to the cells that are producing SPARCL1. Fig1 and S1 are not convincing enough to identify the cells that are producing the protein. Of course, the protein is at least produced by capillary ECs and VE-Cad expressing cells (as shown by the models afterwards), however, it would have been nice to use flow cytometry and intracellular staining for SPARCL1 in order to quantify the importance of ECs in SPARCL1 production at steady state and upon infection. Enlargement of the pictures presented in those figures would also be necessary. Finally, the regions that are enlarged should be shown on the lower magnification pictures.

We appreciate the Reviewer's suggestions; we have provided intracellular flow cytometry data for endothelial cell SPARCL1 as per the Reviewer request. Additionally, we have made modifications to the SPARCL1 immunostaining images as requested. For details, please refer to **Fig. 1G** and **Supplementary Fig. 2B-J**.

- Related to this latter point, the authors should provide an explanation for the fact that, at the mRNA level, SPARCL1 is still produced a lot by ECs at 30 days post infection (and the two subtypes are still very well present). However, at the protein level, SPARCL1 production in BALF is very much reduced. Could it suggest that other cells are the main producers of SPARCL1? Accordingly, mice in which ECs do not or overexpress SPARCL1 display different phenotypes but they are not dramatically different suggesting that SPARCL1 could be produced by other cells that would drive most of the effect.

We appreciate these valuable questions raised by the Reviewer. Regarding the persistent detection of endothelial cell *Sparcl1* mRNA levels at the late stage of infection (day 27), coupled with a significant decrease of SPARCL protein in the BALF, we believe that this is largely explained by the gradual restoration of lung endothelial barrier function. With the completion of tissue repair, the integrity of blood vessels and lung tissue structure is restored, leading to limited leakage of SPARCL1 into the alveoli. We have multiple pieces of evidence supporting this: *a.* After 60 days of infection, serum retains high concentrations of SPARCL1, while there is no significant difference in BALF compared to the homeostasis state (**Suppl Fig. 2K-L**). *b.* During homeostasis, there is no significant difference in SPARCL1 concentration in BALF between WT and endothelial SPARCL1-KO mice. However, a significant decrease is observed in EC^{Sparcl1-KO} mouse lung BALF on day 12 post-injury (**Suppl Fig. 3D-E**). We also added similar description sentences to both *Results* and *Discussion* sections.

Regarding the Reviewer's mention of the observed lack of dramatic differences in mouse phenotypes, firstly, we agree that SPARCL1 from other cell sources might be also involved (e.g., stromal cells, **Suppl Fig. 2A-G**). This is particularly likely to be the case in our EC *Sparcl1*-KO mice, where there may be a supplementary contribution from stromal cell-derived SPARCL1. However, we believe this supplementation is limited, with the primary source still being endothelial cells, especially during the infection period. This was supported by SPARCL1 levels in BALF in WT and EC^{Sparcl1-KO} mice (**Suppl Fig. 3D-E**).

Notably, there is significant variability among mice in viral infections, leading to a large variance in data values, making it challenging to obtain extremely significant differences in results. Nevertheless, we fully agree that SPARCL1 expression from the mesenchyme is worthy of future investigation outside of the scope of this already very data-heavy manuscript.

- The authors show differences at the transcriptomic level until d30 post-infection but do not test later time points. Does it come back to steady state levels? Does the Immu.ECs disappear? Do epigenetic marks persist in those cells?

We appreciate these suggestions. Intriguingly, we have observed a substantial persistence of ECs expressing SPARCL1 even up to 60 days post-infection in the lungs (**Suppl Fig.2H and J**), albeit with a certain degree of reduction. This suggests that this cell population may persist, possibly due to the lungs being in a prolonged inflammatory state after influenza damage. We speculate that these cells may exhibit some epigenetic differences. However, due to the lack of suitable antibodies/surface markers for sorting out this cell population for epigenetic sequencing, this aspect might be addressed in future research. The focus of this study is on elucidating the impact of endothelial SPARCL1 molecules on the local immune microenvironment rather than specifically investigating this cell population. We hope the reviewer can appreciate our perspective on this point.

- It seems that the gating strategies presented in Suppl Fig. 3 and 4 could be improved:

o Some cells of the lung are very autofluorescent. Especially, alveolar macrophages. It therefore appears that the gate for live cells should be increased to the CD45+ APC-Cy7 dim cells that are likely autofluorescent cells. Alternatively, a FMO control for live/dead should be provided.

Thank you for the Reviewer's suggestions. We have made changes as per Reviewer's request, now as new **Suppl Fig.6A**.

o The study highlights an effect of SPARCL1 secretion by ECs of macrophages but do not identify monocytes at any point. This is a major limitation of the paper as those cells are very important upon viral infections (Maquet et al. doi: 10.1126/sciimmunol.abn3240; Li et al. doi: 10.1126/sciimmunol.abj5761). Moreover, it identifies two populations of myeloid cells/macrophages, namely alveolar macrophages and interstitial/recruited macrophages. Firstly, the gating strategy is not shown, however, this is central to the paper (some SiglecF stainings are shown later but a clear gating strategy with all the markers including SiglecF, F4/80, CD11c, CD11b and CD64 is needed). Secondly, this dichotomy is not precise enough. At least, AMs, IMs and monocytes should be distinguished. However, IMs are themselves composed of several subsets that could improve the message of the paper (See Schyns et al, doi: 10.1038/s41467-019-11843-0 or Chakarov et al. doi: 10.1126/science.aau0964). Besides, inflammatory DCs are also cells derived from monocytes that play an important role upon viral infection into the lung (Bosteels et al., doi: 10.1016/j.immuni.2020.04.005), identifying those cells would also be very important. Those

different populations should be clearly distinguished and the related graphs should be modified accordingly.

Thank you for the Reviewer's suggestions. We optimized our gating strategy, incorporating antibodies such as CD11b, CD11c, CD64, F4/80, SiglecF, and others. We assessed changes in the quantities of major myeloid cell populations, including alveolar macrophages (AMs), interstitial macrophages (IMs), inflammatory monocytes, dendritic cells (DCs), and inflammatory dendritic cells (iDCs), during infection under conditions of endothelial *Sparcl1* deficiency or overexpression (**Suppl Fig.4 and Suppl Fig.8B**). While we do understand and appreciate that macrophages and monocytes can be further subsetted, we believe that the take home message, that SPARCL1 promotes a pro-inflammatory macrophage phenotype, is strongly supported by our original and now revised data.

o In the Figure S5, why not taking CD11b+SiglecF+ cells that are recruited cells in differentiation to AMs? In Figure S5, it seems that the gate on F4/80 FMO should be more on the left side as it discards a lot of cells of interest (recruited macrophages) as located

We apologize for the confusion, there was an error in our figure labeling. The corrected X-axis should be Ly6C, where our intention is to exclude the influence of monocytes. This has been rectified. Additionally, regarding the gating strategy for F4/80, we primarily outlined the CD64+F4/80+ macrophage population based on the F4/80 FMO control.

o The authors identify and quantify ILCs which are important but rather small populations but do not quantify neutrophils that are directly related to the severity of viral infections. Here too, several subsets could be identified and the effect of SPARCL1 on those cells should be addressed.

Thank you for pointing that out; we have now added the neutrophil data. (**Suppl Fig.6B and 8A**)

o Figures 4 and S5 compare the expression of CD86 and CD206 in different populations. This could be interesting. However, stainings are not convincing, especially, CD206 staining should be improved. The separation of the different subsets of cells based on CD206 appears as a weakness of the paper. But this is central for the actual distinction between "M1" and "M2" macrophages.

Thank you for the Reviewer's comments. We agree that the flow cytometry gating strategy for CD206 may not appear perfect, given the complexity of a multicolor panel and the intracellular staining of CD206 flow, leading to overlapping cell populations. However, we have implemented a stringent Isotype Control as a reference, and we believe that the CD206 gating strategy is valid. To address this potential limitation, we have conducted extra experiments to quantify the M2-like macrophages, such as RELM α , using both immunostaining and intracellular flow cytometry analysis (**Fig.4D-F**). Further, we did perform RNA-Seq on macrophages from these mice that clearly indicates a SPARCL1 dependent effect on macrophage polarization, buttressing this finding even while recognizing that many flow markers are imperfect.

- The effect of SPARCL1 on other cells is very interesting. However, the sole dichotomy between M1 and M2 orientated cells is too simple. Ideally, a single cell experiment on myeloid cells at day 12 in WT, EC-Sparcl1-KO and EC-SPARKL1-OE would be very informative. Alternatively, showing expression of key regulatory markers (such as PDL1 or CXCL9 (Maquet al. doi: 10.1126/sciimmunol.abn3240) and others) by myeloid cells could improve a lot the message.

We have added additional data as requested. We did not observe any changes in the numbers of myeloid cells expressing CXCL-9 or PD-L1 due to the loss or overexpression of endothelial Sparcl1 during injury (**Suppl Fig.5 and 8**).

- The absence of SPARCL1 effect on populations of myeloid cells at steady state (Fig. S5B) is explained as a consequence of the fact that SPARCL1 fails to infiltrate the alveolar space in uninfected mice. This is not clear. Indeed, only AMs are in the alveolae. Moreover, could it not be that only recruited cells display plasticity and not resident ones at steady state? This should be at least discussed.

These are very insightful questions. Regarding the impact of SPARCL1 on macrophages in the uninjured lung, please refer to the answer to *Question #3*. We agree that in the injured lung, SPARCL1 primarily affects alveolar macrophages, especially the recruited macrophages (as they constitute the majority during injury). Interestingly, we found that SPARCL1 influences both the quantity of recruited macrophages (**Fig.2L, 3N**) and the concentration of CCL-2 in BALF (**Fig.2K and 3M**). This reinforced our model that SPARCL1 contributes to pushing macrophages toward a pro-inflammatory state through NF- κ B activation. As CCL-2 is downstream of NF- κ B, our hypothesis is that SPARCL1 encourages macrophages to release increased CCL-2, attracting monocytes/macrophages and potentially steering them toward a pro-inflammatory phenotype, promoting a feed-forward mechanism of uncontrolled inflammation (**Suppl Fig.17**).

- The authors show that there is no difference in the percentages of different cells (T, B, ...) between the different groups they test (WT, EC-Sparcl1-KO and EC-SPARKL1-OE). However, they do not analyze if those cells exhibit different functional properties (regulatory vs cytotoxic for example). This should be at least discussed.

We have included this part in Discussion section. Thanks to the Reviewer for the suggestions.

- The authors suggest that SPARCL1 acts through TLR4. If it has an effect on recruited monocyte-derived cells, the ideal model would be CCR2CreERT2 TLR4flox mice. This could be tested in order to be fully convincing. This could allow to specifically block TLR4 signalling (and therefore SPARCL1 effect if this is the pathway involved) at precise time points.

We agree that to validate whether *in vivo* endothelial overexpression of *Sparcl1* induces macrophages to shift towards a pro-inflammatory state through TLR4, it would be necessary to utilize macrophage-specific knockout of TLR4 while simultaneously overexpressing SPARCL1 in endothelial cells. However, due to technical challenges associated with

achieving cell-specific overexpression and knockout using the “Cre-loxP” system, and the lack of commercially available “CCR2-DreERT; TLR-4 rox” mice in the current context of “Dre-rox” system development, we find this approach challenging. This was also highlighted by Reviewer 2. In light of these challenges, we have added this discussion to the “Limitations of the study” section.

Minor comments

- In the introduction, the increased expression of SPARCL1 after influenza infection is presented as a fact but no reference is provided. Moreover, this result is presented in the result section. This is not clear and should therefore be modified.

Thank Reviewer for pointing that out. This conclusion is actually a result of our current study, so we have removed this sentence here.

- The initial single cell experiment should be described in a dedicated chapter in the material and methods. Indeed, at the moment, it appears complicated to know the number of mice that were used, their sex and the dose of virus with which they have been infected.

We have added more details about the animals and treatments in this part as requested. Thank you for bringing this to our attention.

- In Figure 1, the EC subsets should be shown as indicated on Figure S1

We have labeled all EC subsets in Fig.1 as suggested.

- Spaces are missing before references at several locations

Thanks for pointing out, we updated all reference format to the Nature Comms style. This issue should be corrected.

- The authors have made several mutant mice, which is rather impressive. For the EC-Sparcl1-KO, what is the PCR used in Figure 2C? Could the authors identify those primers on Figure 2A?

We already put these primers and PCR program in **Table S2**.

- The authors measure different cytokines that are interesting to measure inflammation but some difference could also relate in the recruitment of monocytes. Therefore, it seems that measurements of BALF and circulating CCL2 could also be interesting.

Thanks for the Reviewer’s suggestion, we have included CCL2 data (**Fig.2K, 3M; Suppl Fig.3I and 7F**) in this revision. We did observe interesting SPARCL1-dependent changes in CCL2, so are particularly appreciative of this suggestion!

- Fig. 4D is not convincing. The difference in RELMa expression is very different but much bigger than the one reported on FACS with identification of M2 macrophages. Quantification of RELMa+ cells by FACS should be performed.

We performed RELM α intracellular flow and added it in Fig.4D-E.

- Fig. 5: in the text, the authors suggest differences of BMDM orientation based on morphology. This is not convincing.

Thank you for the suggestion. We have moved this panel to the Suppl Fig.15.

- In the discussion, the authors talk about differences in Th1 inflammation as a fact, however, this is not tested.

We have removed all instances of "Th1" throughout the manuscript and replaced them with "pro-inflammatory" or similar terms.

REVIEWERS' COMMENTS

Reviewer #1 (Remarks to the Author):

My concerns have been addressed in the manuscript revisions submitted by the author. Congratulations on this work!

Reviewer #2 (Remarks to the Author):

The authors have significantly improved the mechanistic studies in the revised manuscript and adequately responded to my comments.

Reviewer #3 (Remarks to the Author):

The authors provided an interesting amount of new data to respond to the reviewers' comments and support their conclusions.

Overall, this has improved the document. Some of the issues relating to the models have not been addressed due to the difficulty of implementing them in a limited timeframe.

The data are interesting and convincing and illustrate how, during pulmonary viral infection, endothelial cells play a central role and can influence the function of certain immune cells such as myeloid cells.

REVIEWERS' COMMENTS

Reviewer #1 (Remarks to the Author):

My concerns have been addressed in the manuscript revisions submitted by the author. Congratulations on this work!

We appreciate all the insightful critiques from this reviewer!

Reviewer #2 (Remarks to the Author):

The authors have significantly improved the mechanistic studies in the revised manuscript and adequately responded to my comments.

We are thrilled to have provided appropriate experimental responses to this reviewer's concerns.

Reviewer #3 (Remarks to the Author):

The authors provided an interesting amount of new data to respond to the reviewers' comments and support their conclusions.

Overall, this has improved the document. Some of the issues relating to the models have not been addressed due to the difficulty of implementing them in a limited timeframe.

The data are interesting and convincing and illustrate how, during pulmonary viral infection, endothelial cells play a central role and can influence the function of certain immune cells such as myeloid cells.

We are thankful for all the reviewer's input and we appreciate their understanding that a few points could not be comprehensively addressed due to technical limitations.